# Incentivizing Truthful Language Models via Peer Elicitation Games

**Baiting Chen**[*]
UCLA
brantchen@ucla.edu

**Tong Zhu**[*]
UCLA
toz015@ucla.edu

**Jiale Han**
UCLA
jialehan@ucla.edu

**Lexin Li**
UC Berkeley
lexinli@berkeley.edu

**Gang Li**
UCLA
vli@ucla.edu

**Xiaowu Dai**[†]
UCLA
daix@ucla.edu

## Abstract

Large Language Models (LLMs) have demonstrated strong generative capabilities but remain prone to inconsistencies and hallucinations. We introduce Peer Elicitation Games (PEG), a training-free, game-theoretic framework for aligning LLMs through a peer elicitation mechanism involving a generator and multiple discriminators instantiated from distinct base models. Discriminators interact in a peer evaluation setting, where utilities are computed using a determinant-based mutual information score that provably incentivizes truthful reporting without requiring ground-truth labels. We establish theoretical guarantees showing that each agent, via online learning, achieves sublinear regret in the sense their cumulative performance approaches that of the best fixed truthful strategy in hindsight. Moreover, we prove last-iterate convergence to a truthful Nash equilibrium, ensuring that the actual policies used by agents converge to stable and truthful behavior over time. Empirical evaluations across multiple benchmarks demonstrate significant improvements in factual accuracy. These results position PEG as a practical approach for eliciting truthful behavior from LLMs without supervision or fine-tuning.

## 1 Introduction

LLMs have achieved remarkable progress in natural language generation, reasoning, and few-shot learning [62, 1, 37, 32]. Despite these advances, they remain fundamentally limited by hallucination—the generation of outputs that are inconsistent with previous responses or factually incorrect [40, 3, 26, 25]. A common failure mode is inconsistency: LLMs may produce different answers to semantically equivalent prompts due to stochastic decoding or sensitivity to minor variations in phrasing [61, 40, 19, 12].Another major challenge is lack of truthfulness, where outputs are not semantically accurate or aligned with verifiable facts, even when the relevant knowledge is implicitly encoded in the model [3, 17]. These limitations are particularly concerning in high-stakes domains such as scientific discovery, education, and decision-making, where outputs must be both consistent and truthful [59, 25]. This motivates a central question: *How can we reliably elicit consistent and truthful behavior from LLMs?*

To address this question, a growing line of work has explored post-training alignment techniques, such as supervised fine-tuning and reinforcement learning with human feedback [4, 52, 54]. While

---

[*]Equal contribution.

[†]*Address for correspondence:* Xiaowu Dai, Department of Statistics and Data Science, UCLA, 8125 Math Sciences Bldg #951554, Los Angeles, CA 90095, USA. Email: daix@ucla.edu.

39th Conference on Neural Information Processing Systems (NeurIPS 2025).

these methods can be effective, they are typically computationally intensive, require extensive human annotation, and lack theoretical guarantees of truthful behavior [31, 47, 52]. Their dependence on model internals also limits their scalability and transferability across different LLMs [9, 61].

An alternative line of research draws on game-theoretic frameworks that aim to improve LLM reliability through structured multi-agent interactions [13, 27]. These approaches offer the advantage of training-free alignment using only black-box access to LLMs. For example, the *consensus game* aligns a generator and a discriminator by rewarding agreement between their outputs [27]. However, because the objective is based on mutual agreement, it can lead to uninformative or collusive equilibria, where agents reinforce each other's responses even when those responses are factually incorrect.

We propose *Peer Elicitation Games* (PEG), a training-free, game-theoretic framework for aligning LLMs through structured peer evaluation. In PEG, a generator produces candidate responses to prompts, and multiple independently instantiated LLMs act as discriminators. Each discriminator assesses the generator's output and is, in turn, evaluated by the other discriminators—who serve as peer referees. This mutual evaluation mechanism assigns utilities based on the level of agreement among discriminators, encouraging truthful reporting without relying on ground-truth labels. This incentive structure ensures that truthful behavior by each discriminator constitutes a Nash equilibrium, while discouraging collusion or uninformative consensus. To implement the framework, we apply the online mirror descent algorithm to iteratively update each discriminator's policy, enabling the system to converge toward equilibrium through repeated, utility-driven interaction. The majority vote among discriminators is then returned to the generator, serving as a feedback signal for improving future generations. The design of PEG—introducing multiple LLM discriminators and rewarding them through peer evaluation—is inspired by renowned concepts in biology, including cognitive synergy [38, 55] and collective intelligence [64, 7], where diverse agents, each holding partial or noisy information, can collectively arrive at judgments that surpass what any individual could achieve alone. PEG enables language models to self-organize toward truthful and stable outputs using only local incentives. This collective dynamic offers a scalable and supervision-free approach to building more trustworthy LLM systems.

To summarize, the main contribution of our work is as follows:

- We propose PEG, a training-free framework for eliciting truthful behavior from LLMs, without relying on ground-truth labels or model fine-tuning. The framework casts the interaction between a generator and multiple heterogeneous discriminators as a multi-agent peer evaluation game as depicted in Figure 1.

- We provide theoretical guarantees showing that PEG promotes truthful reporting as a Nash equilibrium. Furthermore, when each discriminator updates its policy using online mirror descent, the system achieves sublinear regret and converges to a truthful Nash equilibrium over repeated interactions.

- Through experiments on a range of benchmarks, including ARC, MMLU, and GPQA, we demonstrate that PEG improves factual accuracy by over 10% compared to existing methods. Additionally, our results show that smaller models (e.g., 7B parameters) using PEG can match or outperform much larger models (e.g., 65B).

**Related Work.** This work relates to several lines of research. *First*, game-theoretic frameworks have been explored to improve reasoning and alignment in LLMs through structured interactions [13]. Most relevant is the Consensus Game framework [27], which promotes self-consistency by reconciling the outputs of a generator LLM and a discriminator LLM through game-theoretic interaction. Follow-up studies extend this perspective to decoding [11], federated learning with competing interests [65], and embodied tasks like vision-language navigation [67]. Our work builds on this foundation by introducing a multi-agent formulation with explicit peer interaction and a utility mechanism that promotes truthful reporting rather than mere agreement. *Second*, a growing body of research demonstrates that multi-agent systems of LLMs can collaborate through debate and cooperation to enhance factuality and task performance [36, 15, 44]. Our approach draws on this idea by coordinating multiple LLMs through structured peer evaluation. *Third*, our work connects to the literature on learning Nash equilibria in multi-agent systems, particularly no-regret learning in general-sum games [24, 60, 8] and preference-based learning frameworks such as Nash learning from human feedback [41] and direct preference optimization [58]. We contribute to this literature by showing how no-regret learning dynamics converge to truthful equilibria in a structured, incentive-compatible setting. *Finally*, PEG builds on peer prediction mechanisms [43, 39], which

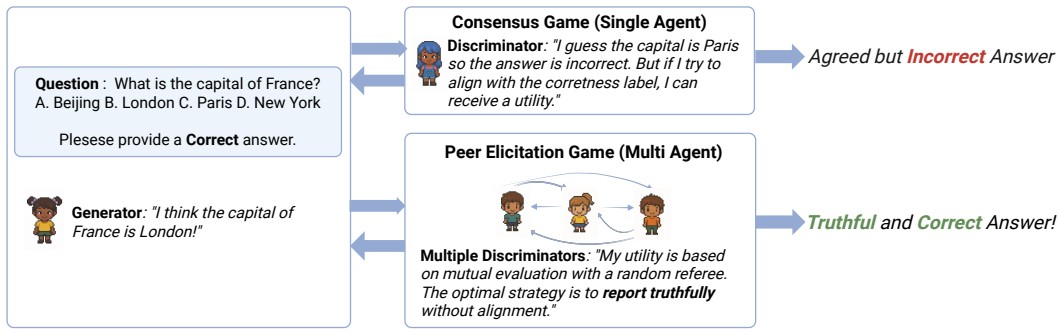

Figure 1: Comparison of the consensus game and PEG: when multiple discriminator LLMs independently evaluate the generator's output and are rewarded based on mutual agreement, their collective judgment aligns more closely with true answers.

have been extended to handle complex settings [63, 46, 68, 49]. We adapt these ideas to LLM agents by extending multi-task peer prediction mechanisms [2, 33] to a setting where multiple LLMs jointly evaluate responses via mutual scoring and incentive-aligned interaction.

## 2 Methodology

This section introduces the key components of PEG including a formal definition of truthfulness and an overview of the interactions between the generator and multiple discriminator agents.

### 2.1 Truthfulness as Incentive Compatibility

We adopt the concept of *incentive compatibility* (IC) from mechanism design [42] to define the truthfulness. IC ensures that each agent maximizes its utility by reporting its true private information. Specifically, let $c_i$ denote the true private information of agent $i$, and $r_i$ a possible report. In our setting, the agents are discriminators, each independently evaluating a generated response. The true private information $c_i \in \{0, 1\}$ refers to the agent's truthful judgment on whether the response is factually correct ($c_i = 1$) or incorrect ($c_i = 0$). The report $r_i \in \{0, 1\}$ is the label that the agent chooses to submit to the system based on (or possibly deviating from) this truthful judgment. Let $c_{-i}$ represent the truthful information of all other agents, The mechanism assigns utility $u_i(r_i, r_{-i})$ to agent $i$ based on its own report and the reports of all other agents. A mechanism is said to be *incentive compatible (IC)* if, for all agents $i$, all $c_i, c_{-i}$, and all possible reports $r_i$,

$$u_i(c_i, c_{-i}) \geq u_i(r_i, c_{-i}). \tag{1}$$

The consensus game framework [27] involves one LLM generator and one LLM discriminator, aiming to improve consistency by rewarding agreement between agents. However, this structure does not guarantee truthfulness, as agents are rewarded only for agreement rather than accuracy and may benefit from reporting false but mutually agreeable outputs. For example, the generator produces an incorrect answer to a question when tasked with generating a correct one. The discriminator, whose truthful evaluation would be to mark the answer as incorrect ($c_i = 0$), may instead receive a higher utility for agreeing with the generator by reporting it as correct ($r_i = 1$). In such a scenario, the utility of truthful reporting is lower than that of misreporting, i.e., $u_i(c_i, c_{-i}) < u_i(r_i, c_{-i})$, which creates a clear incentive for the discriminator to misreport. This violates the IC property defined in Eq. (1), highlighting a fundamental limitation of consensus-based approaches..

In contrast, recent studies have shown that multi-agent debates can better integrate the diverse perspectives of multiple models, leading to more accurate and reliable outputs [10, 15]. Motivated by these findings, we propose PEG in which multiple discriminator agents independently evaluate each generated response. In this setup, each discriminator is rewarded based on agreement with its peer discriminators. We show that when all other discriminators report truthfully, the best response for any individual discriminator is also to report truthfully—thus, truthful reporting becomes a Nash equilibrium. That is, PEG achieves IC property defined in (1), where no agent can improve its utility

by deviating unilaterally. Figure 1 illustrates a key distinction between our PEG and the consensus game, where the generator and a single discriminator are incentivized to align with each other, which can lead to consistent but incorrect outputs. In contrast, PEG relies on independent mutual evaluations from multiple discriminators, promoting outputs that are not only consistent but also correct outputs.

## 2.2 Peer Elicitation Games (PEG)

We consider a generator $G$ and a set of $n$ discriminators $D_1, D_2, \ldots, D_n$. Each generator and discriminator maintains a probabilistic policy. At each round $t \in \{1, 2, \ldots, T\}$, the system assigns a set of tasks indexed by $k_t = 1, 2, \ldots, K_t$, where $K_t$ is the number of tasks in round $t$. For each task $(t, k_t)$, the generator receives an input question $X_{t,k_t}$ along with a correctness label $V_{t,k_t} \in \{0, 1\}$, and produces a response $Y_{t,k_t}$ according to its probabilistic policy. The generator's policy is a conditional distribution over responses given the input and target label:

$$\pi_G(Y_{t,k_t} \mid X_{t,k_t}, V_{t,k_t}) = \mathbb{P}(Y_{t,k_t} \mid X_{t,k_t}, V_{t,k_t}; \theta_G),$$

where $\theta_G$ denotes the generator's parameters.

Each discriminator $D_i$ observes $X_{t,k_t}$ and the corresponding generator response $Y_{t,k_t}$, and outputs a predicted label $V_{i,t,k_t} \in \{0, 1\}$ according to its policy:

$$\pi_{D_i}(V_{i,t,k_t} \mid X_{t,k_t}, Y_{t,k_t}) = \mathbb{P}(V_{i,t,k_t} \mid X_{t,k_t}, Y_{t,k_t}; \phi_i),$$

where $\phi_i$ denotes the parameters of discriminator $D_i$.

The goal is to generate responses that are both consistent and truthful. Our method achieves this by aligning the generator's output with the majority judgment of discriminators, while incentivizing truthful evaluation from discriminators through peer evaluation. The overview of our method is illustrated in Figure 2. The left branch (in red) corresponds to the supervised alignment goal of imitating majority vote labels, while the right branch (in blue) depicts the PEG that ensures incentive compatibility among discriminators.

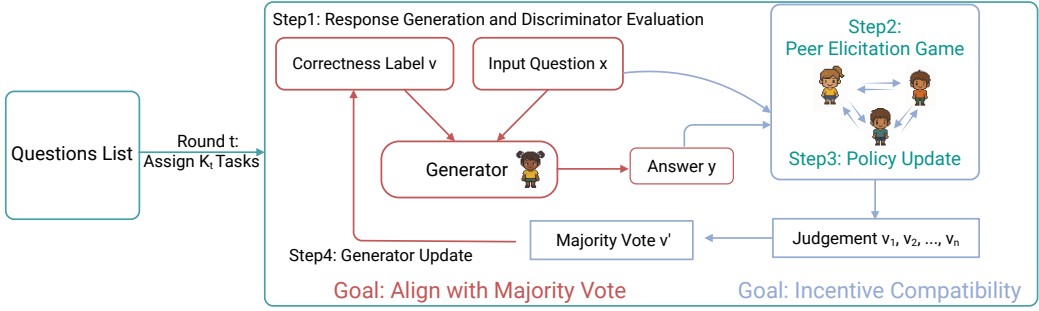

Figure 2: Overview of our method: multiple discriminators independently evaluate the response provided by the generator, while each discriminator is rewarded based on mutual agreement with peers via PEG. This setup incentivizes truthful reporting for discriminators and aligns the generator without requiring ground-truth labels.

**Step 1: Response Generation and Discriminator Evaluation.** At each round, given an input question $X_{t,k_t}$ and a correctness label $V_{t,k_t}$, the generator produces a response $Y_{t,k_t} \sim \pi_G(\cdot | X_{t,k_t}, V_{t,k_t})$. Each discriminator independently evaluates the response based on the question and outputs a correctness report $V_{i,t,k_t} \sim \pi_{D_i}(\cdot | X_{t,k_t}, Y_{t,k_t})$.

**Step 2: Peer Elicitation Games.** To incentivize truthful reporting, the discriminator agents engage in a peer elicitation game, where the utility of each agent is based on a mutual evaluation. Importantly, this utility requires only reports from discriminators and does not rely on access to ground truth. Assume that in round $t$, all discriminators are assigned $K_t$ tasks. For each task $k_t \in [K_t]$, discriminator $i$ privately observes a signal $C_{i,t,k_t} \in \{0, 1\}$ and submits a report $V_{i,t,k_t} \in \{0, 1\}$. The task set $\{1, \ldots, K_t\}$ is arbitrarily partitioned into two disjoint subsets $\mathcal{K}_{1,t}$ and $\mathcal{K}_{2,t}$. For every pair of distinct agents $i \neq j \in [n]$, we construct two $2 \times 2$ co-report matrices $\mathbf{M}_{1,t}^{ij}$ and $\mathbf{M}_{2,t}^{ij}$, one for

each subset $\mathcal{K}_{\ell,t}$ where $\ell = 1, 2$. Each entry of $\mathbf{M}_{\ell,t}^{ij}(c, c')$ counts the number of tasks $k_t \in \mathcal{K}_{\ell,t}$ for which agents $i$ and $j$ reported the pair $(c, c')$, i.e.,

$$\mathbf{M}_{\ell,t}^{ij}(c, c') := \sum_{k_t \in \mathcal{K}_{\ell,t}} \mathbb{1}\big((V_{i,t,k_t}, V_{j,t,k_t}) = (c, c')\big).$$

The total payment to agent $i$ in round $t$ is defined as the sum over all other agents of the product of the determinants of these matrices:

$$p_{i,t} := \sum_{j \neq i} \det(\mathbf{M}_{1,t}^{ij}) \cdot \det(\mathbf{M}_{2,t}^{ij}).$$

In our setting, the discriminators are rewarded based on mutual evaluation. Meanwhile, the generator is incentivized to produce responses that align with the majority consensus among high-confidence discriminators. The utility functions are computed over a batch of $K_t$ tasks in round $t$. The overall utility functions are defined as:

$$
\begin{aligned}
u_{D_i}^{(t)} &= \mathbb{E}_{V_{i,t,k_t} \sim \pi_{D_i}(\cdot | X_{t,k_t}, Y_{t,k_t})} [p_{i,t}], \quad \forall i \in \{1, \dots, n\}, \\
u_G^{(t)} &= \mathbb{E}_{Y_{t,k_t} \sim \pi_G(\cdot | X_{t,k_t}, V_{t,k_t})} \left[ \mathbb{1}(V_{t,k_t} = \hat{V}_{t,k_t}) \right],
\end{aligned}
\tag{2}
$$

where $k_t \in \{1, \dots, K_t\}$ indexes the tasks in round $t$, and $\hat{V}_{t,k_t}$ denotes the majority vote label aggregated from the discriminator reports on task $(t, k_t)$.

**Step 3: Policy Update.** We assume that the policy space consists of conditional probability distributions over inputs, and focus on a localized subset around a reference policy $\pi^*$ (e.g., a truthful reporting strategy) [50]. This constraint reflects the intuition that although policies may adapt to optimize for high utility, they should not obviate too far from truthful behavior to maintain semantic consistency and interpretability. In practical terms, this can be enforced by initializing the agent with a fine-tuned model that embodies $\pi^*$, and constraining updates to remain within a trust region [45, 69]. Specifically, we define the local policy neighborhood as:

$$\Pi_{\text{local}} = \{\pi(\cdot \mid \text{input}; \theta) \in \Pi \mid \|\pi(\cdot \mid \text{input}; \theta) - \pi^*(\cdot \mid \text{input})\| < \delta\}. \tag{3}$$

Our goal is to iteratively learn the policy that maximizes the utility function, aligning with the online learning framework that sequentially optimizes an objective function by searching for its critical point [22, 51]. Since each policy is a probability distribution, we adopt the Online Mirror Descent (OMD) algorithm with negative entropy as the Bregman divergence, which naturally preserves the probabilistic structure of the policy and regularizes each update by penalizing large deviations from the previous policy [16]. The update rules for the discriminators and generator are:

$$
\begin{aligned}
\pi_{D_i}^{(t'+1)}(V_{i,t,k_t} \mid X_{t,k_t}, Y_{t,k_t}) &\propto \pi_{D_i}^{(t')}(V_{i,t,k_t} \mid X_{t,k_t}, Y_{t,k_t}) \exp\left[ \eta_t^{D_i} \nabla_{\pi_{D_i}^{(t')}(V_{i,t,k_t}|X_{t,k_t},Y_{t,k_t})} u_{D_i}^{(t')} \right], \\
\pi_G^{(t'+1)}(Y_{t,k_t} \mid X_{t,k_t}, V_{t,k_t}) &\propto \pi_G^{(t')}(Y_{t,k_t} \mid X_{t,k_t}, V_{t,k_t}) \exp\left[ \eta_t^G \nabla_{\pi_G^{(t')}(Y_{t,k_t}|X_{t,k_t},V_{t,k_t})} u_G^{(t')} \right].
\end{aligned}
\tag{4}
$$

Intuitively, these updates guide each agent to adjust its policy to direction yields higher utility. After updating, the individual judgments $\{V_{1,t,k_t}, \dots, V_{n,t,k_t}\}$ from all discriminators on task $(t, k_t)$ are aggregated via majority vote to form a consensus label $\hat{V}_{t,k_t}$, which serves as a proxy for correctness.

**Step 4: Generator Update.** The utility of the generator is determined by whether its output aligns with the consensus label derived from the discriminators. Specifically, the generator receives utilies when its generated response matches the consensus label $\hat{V}_{t,k_t}$. This design enables learning without requiring supervised fine-tuning or access to explicit ground-truth correctness labels.

## 3 Theoretical Guarantees

In this section, we present three main theoretical results: (i) the mechanism incentivizes dominantly truthful reporting in Section 3.1; (ii) both the generator and discriminators achieve sublinear regret under online learning dynamics in Section 3.2; and (iii) last-iterate converges to a truthful Nash equilibrium in Section 3.3. Here, last-iterate convergence means that the policy used in the final iteration converges to the equilibrium, rather than requiring averaging over past iterations [6].

## 3.1 Dominant Truthfulness

PEG satisfies the dominant truthfulness property, meaning that truthful reporting is a dominant strategy for each discriminator, as it yields the highest expected utility regardless of the strategies chosen by other agents. This ensures that truthful behavior is consistently incentivized. We now formally state this property of PEG.

**Lemma 1.** *Let $n$ be the number of agents (e.g., discriminators) and $K_t$ be the number of tasks assigned in round $t$ as defined in Section 2.2. When $n \geq 2$ and $K_t \geq 4$, under mild assumptions, PEG is dominantly truthful and satisfies IC in Eq. (1). That is, for every agent $i$, the truthful reporting strategy maximizes their expected payment regardless of the strategies chosen by other agents.*

The proof of this lemma, which leverages ideas from [33], is provided in Appendix A.1. The design of PEG is grounded in three principles to ensure this incentive guarantee. First, the use of determinant-based utility ensures information-monotonicity: the determinant achieves its maximum value when agents report truthfully, making truthful reporting the most rewarding strategy. Second, the utility function acts as an unbiased estimator of the joint distribution of agent reports, allowing the mechanism to approximate the true distribution without estimation error. Finally, to prevent negative payments, the set of tasks is divided into two disjoint subsets, and payments are computed using the product of determinants from these subsets. As both subsets serve as unbiased estimators of the distribution of agent reports, their determinants are expected to have the same sign. As a result, the payment is always non-negative.

## 3.2 Regret Analysis

Next, we show that both the generator and the discriminators can progressively improve their behavior such that their cumulative performance asymptotically approaches that of the best truthful policy by performing online policy updates as defined in Eq. (4) in Section 2.2. This is formalized via the standard notion of *no-regret learning*, which measures the cumulative difference between the utility obtained by the learned policy over time and the utility that would have been achieved by the best fixed policy in hindsight [6, 18]. Since the generator and discriminators are updated independently, and their optimal strategy is to report truthfully regardless of others, we follow the setting in [27] and define the regret for each discriminator and the generator as:

$$\text{Regret}_{D_i}(T) = \sum_{t'=1}^{T} u_{D_i}^{(t,t')}(\pi_{D_i}^{(t')}) - \max_{\pi_{D_i}^*} \sum_{t=1}^{T} u_{D_i}^{(t,t')}(\pi_{D_i}^*),$$

$$\text{Regret}_{G}(T) = \sum_{t=1}^{T} \sum_{k_t=1}^{K_t} u_{G}^{(t,t',k_t)}(\pi_{G}^{(t')}) - \max_{\pi_{G}^*} \sum_{t'=1}^{T} \sum_{k_t=1}^{K_t} u_{G}^{(t,t',k_t)}(\pi_{G}^*),$$

where $\pi_{D_i}^{(t')}$ and $\pi_{G}^{(t')}$ denote the policies of discriminator $D_i$ and the generator at iteration $t'$, and $\pi_{D_i}^*, \pi_{G}^*$ are their respective best fixed policies in hindsight. Each $u_{G}^{(t,k_t)}(\cdot)$ denotes the generator's utility on task $k_t$ in round $t$.

We introduce the following Assumption 1 for theoretical guarantees.

**Assumption 1.** *The utility function $u_{D_i}(\pi_{D_i})$ and $u_G(\pi_G)$ satisfies the following:*

*(Part 1: Local concavity). There exists a neighborhood $\mathcal{N}$ around a truthful reference policy such that $u_{D_i}(\pi_{D_i})$ and $u_G(\pi_G)$ is concave in $\pi_{D_i}$ and $\pi_G$ for all $\pi_{D_i}, \pi_G \in \mathcal{N}$.*

*(Part 2: Gradient boundedness). There exist constants $M_1, M_2 > 0$ such that for all $\pi_{D_i}$ and $\pi_G$, the gradients are bounded in $\ell_2$-norm: $\|\nabla_{\pi_{D_i}} u_{D_i}(\pi_{D_i})\|_2 \leq M_1$, and $\|\nabla_{\pi_G} u_G(\pi_G)\|_2 \leq M_2$.*

Part 1 of Assumption 1 assumes that the utility function is locally concave with respect to the policy within a restricted neighborhood around the truthful policy $\pi^*$, as defined in Section 2.2. This is justified because $\pi^*$ is designed to be utility-maximizing under PEG, and policy updates are constrained to stay close to $\pi^*$ in practice via deploying a fine-tuned model. Similar locality and curvature assumptions are standard in trust-region and online learning methods [50, 20]. Part 2 of Assumption 1 requires that the gradients of the utility function are bounded. These assumptions are standard in the convex optimization literature such as [6, 16] which also employ gradient-based optimization methods.

We now formalize the no-regret property of PEG. The following theorem shows that both the generator and each discriminator achieve sublinear regret when updated via mirror descent with appropriately chosen learning rates.

**Theorem 1.** *Under Assumption 1, and set the learning rate in Eq. (4) for the generator as $\eta^G := \sqrt{2D_{\mathrm{KL}}(\pi_G^* \| \pi_G^{(1)})/(M_1^2 t_l)}$, and for each discriminator $D_i$ as $\eta^{D_i} := \sqrt{2D_{\mathrm{KL}}(\pi_{D_i}^* \| \pi_{D_i}^{(1)})/(M_2^2 t_l)}$ for $2^{t_l} \leq T < 2^{t_l+1}$. Then, the regrets of the generator and each discriminator are bounded by:* $\mathrm{Regret}_G(T) \leq \frac{\sqrt{2}}{\sqrt{2}-1} M_1 \sqrt{2K_T D_{\mathrm{KL}}(\pi_G^* \| \pi_G^{(1)}) \cdot T}$, *and* $\mathrm{Regret}_{D_i}(T) \leq \frac{\sqrt{2}}{\sqrt{2}-1} M_2 \sqrt{2K_T D_{\mathrm{KL}}(\pi_{D_i}^* \| \pi_{D_i}^{(1)}) \cdot T}$, *respectively, where $D_{\mathrm{KL}}(\cdot \| \cdot)$ denotes the KL divergence between the optimal policy $\pi^*$ and the initial policy $\pi^{(1)}$, and $K_T$ is the number of tasks at $T$ iteration.*

Theorem 1 guarantees that both the generator and each discriminator achieve a regret bound of $O(\sqrt{T})$, which is consistent with standard results in online convex optimization and learning theory [e.g., 6, 30]. This sublinear regret implies that the policy update defined in Eq. (4) is *Hannan consistent*: their average regret vanishes as $T \to \infty$, meaning that each agent's average performance converges to that of the best fixed policy [28]. More specifically, the regret bound depends on three key factors: (1) the bound on the gradient norm of the utility functions; (2) the number of tasks $K$; and (3) the KL divergence between the initial policy and the optimal one, which measures how far the agent's starting point is from the target behavior. A smaller KL divergence leads to a tighter regret bound, as the learning trajectory begins closer to the optimal policy. Importantly, regret analysis does not imply convergence to the optimal policy, but ensures that the average utility gap to the best fixed policy vanishes as $T \to \infty$. While regret guarantees are well studied in online learning, applying them to a peer evaluation setting with multiple interdependent LLMs is novel. In this setting, agents influence each other only indirectly through their reported outputs, making this a theoretically grounded and promising direction for aligning LLMs.

### 3.3 Last-Iterate Convergence

Our third main theoretical result concerns the convergence behavior of agents to a Nash equilibrium. Specifically, we establish last-iterate convergence, a stronger guarantee that ensures that the actual sequence of policies converges to a fixed point [6].

In our PEG setup, each discriminator interacts with others repeatedly, aiming to report truthful evaluations based on shared signals. These interactions can be naturally modeled as a continuous multi-agent game $\mathcal{G} = (\mathcal{N}, \Pi, \{u_i\}_{i \in \mathcal{N}})$, where $\mathcal{N} = \{1, \ldots, n\}$ denotes the set of agents, $\Pi = \prod_{i=1}^{n} \Pi_i$ is the joint space of stochastic policies, and $u_i : \Pi \to \mathbb{R}$ is the utility function for agent $i$, reflecting agreement with peers. We assume each $u_i$ is continuous, differentiable in its own argument, and has Lipschitz continuous gradients.

A Nash equilibrium in this setting corresponds to a stable configuration of policies where no discriminator has an incentive to unilaterally deviate [23, 34]. Formally, a joint policy $\pi^* = (\pi_1^*, \ldots, \pi_n^*) \in \Pi$ is a Nash equilibrium if for every $i \in \mathcal{N}$ and any alternative policy $\hat{\pi}_i \in \Pi_i$,

$$u_i(\pi_i^*, \pi_{-i}^*) \geq u_i(\hat{\pi}_i, \pi_{-i}^*). \tag{5}$$

**Theorem 2** (Last-iterate Convergence to Nash). *Suppose each discriminator $i \in \mathcal{N}$ updates its policy using Eq. (4) with a decaying learning rate of order $O(1/t^p)$, for some $p \in (\frac{1}{2}, 1)$. Then, the sequence of joint policies $\pi_t$ converges almost surely to the unique Nash equilibrium $\pi^*$ of PEG.*

The proof of Theorem 2, provided in Appendix A.3, builds on the Robbins–Siegmund Lemma. It is also inspired by the analysis in [16], though our setting is more specific—focusing on static games with perfect feedback. Theorem 2 establishes last-iterate convergence to the Nash equilibrium: the actual policies used by each discriminator at the end of training converge to the truthful equilibrium. This is in contrast to classical no-regret learning, which only ensures good average performance over time. Last-iterate convergence is particularly valuable in practice, as it guarantees that the learned policies are not just good on average but are inherently stable and reliable [6, 16]. In our case, this means PEG reliably leads to consistent and truthful evaluations, effectively enabling smaller language models to coordinate and reach human-aligned consensus without any supervised fine-tuning or distillation.

Table 1: Accuracy (%) of majority vote answers across benchmark datasets for each method. **Bold** indicates the best performance in each row.

| Dataset | G | D | MI | ER-D | PEG |
|---|---|---|---|---|---|
| ARC-Easy | 88.19 | 84.18 | 88.61 | 88.57 | **91.78** |
| ARC-Challenge | 77.01 | 70.68 | 78.03 | 77.52 | **87.01** |
| MMLU | 59.98 | 50.75 | 59.85 | 59.66 | **70.73** |
| GPQA-Main | 18.08 | 9.15 | 16.29 | 16.52 | **22.54** |

## 4 Experiments

We evaluate PEG on question-answering (QA) tasks. In this setup, a generator LLM agent produces answers, while a group of discriminator LLM agents independently evaluate each response, serving as peers to promote both accuracy and consistency. Code for all experiments is available at `https://github.com/toz015/neurips2025-repo`.

### 4.1 Experiment Setup

**Models.** The main experiment employs the following models: DeepSeek-R1-Distill-Qwen-7B (deepseek-Qwen-7b), deepseek-ai/deepseek-llm-7b-chat (deepseek-Llama-7b), and Qwen/Qwen2.5-7B-Instruct (OQwen-7b) as discriminators, with Qwen/Qwen2.5-7B-Instruct also serving as the generator. We consider Gemma-7B [53], Mistral-7B [29], Ai-Yi-9B [66] and OpenChat-7B [57] as candidate discriminators. Unless otherwise specified, we set the learning rate $\eta = 0.1$ for all experiments. The PEG mechanism between discriminators is run for 10 iterations for 8 tasks. Discussions on different choices of learning rates and number of iterations are provided in Appendix C.6 and C.7.

**Prompts.** We evaluate our models with zero-shoting following the format described in [21]. By default, conditioning the $P_{LM}$ on $(x, \text{correct})$ corresponds to the standard zero-shot prompt. Conditioning on $(x, \text{incorrect})$ uses the same structure, except that the phrase "Answer:" is replaced with "Incorrect Answer:" in the prompt C.1.

**Baselines.** We incorporate several baseline methods, each representing a different approach to generating and selecting responses. To evaluate against our method, we apply each baseline to obtain responses from the discriminators, and then compare their outputs using majority voting [61].

- **Generative Ranking (G)**: A standard baseline that ranks candidate answers by the probability $P_{LM}(y \mid x, \text{correct})$ and selects the top candidate [56].
- **Discriminative Ranking (D)**: This method employs a discriminator $\pi_D$ to estimate $P(\text{correct} \mid x, y)$ and ranks responses accordingly [27].
- **Mutual Information Ranking (MI)**: This method reweights each candidate by the product of the forward and reverse likelihoods, $P_{LM}(y \mid x, \text{correct}) \cdot P_{LM}(\text{correct} \mid x, y)$ [35].
- **Equilibrium Ranking Discriminator (ER-D)**: Based on the Consensus Game framework of [27], this method formulates the interaction between a generator and a discriminator as a signaling game. The discriminator iteratively updates its estimates to maximize agreement with the generator. Each query–candidate pair $(x, y)$ is reweighted by $\pi_D^*(\text{correct} \mid x, y)$, encouraging consistency in the final ranking.

**Datasets.** We conduct our evaluations using four diverse datasets: ARC-Easy, ARC-Challenge, Massive Multitask Language Understanding (MMLU) and Graduate-Level Google-Proof Q&A Benchmark (GPQA) [21, 14, 48]. Details of the datasets are in Appendix C.2. Each dataset presents unique challenges, allowing us to test PEG under various knowledge domains.

### 4.2 Experiment Results

**PEG Improves Accuracy.** Results in Table 1 show that PEG consistently outperforms all baselines across the evaluated datasets. Notably, it achieves more than a 10% improvement in accuracy on the most challenging benchmarks, ARC-Challenge and MMLU, compared to the strongest baseline. This performance gain is due to two key factors. First, PEG leverages the complementary strengths and

Table 2: Accuracy (%) of each model before (D) and after applying the PEG mechanism (PEG) across four benchmark datasets. **Bold** highlights the best PEG result per dataset.

| Model | ARC-Easy | | ARC-Challenge | | MMLU | | GPQA | |
| --- | --- | --- | --- | --- | --- | --- | --- | --- |
| | D | PEG | D | PEG | D | PEG | D | PEG |
| OQwen-7B | 90.43 | 90.89 | 82.91 | 85.04 | 62.72 | 68.19 | 16.07 | 22.77 |
| deepseek-Qwen-7B | 70.48 | **91.73** | 57.18 | **86.15** | 42.38 | **69.04** | 15.18 | 20.09 |
| deepseek-LLaMA-7B | 76.30 | 91.44 | 59.83 | 85.98 | 46.23 | 68.71 | 17.41 | **23.66** |

diverse reasoning capabilities of multiple LLM discriminators. We provide an illustrative example in Appendix 6a to further validate that initial divergent outputs will become more consistent and accurate after applying PEG. Second, unlike agreement-based methods such as MI and ER-D, PEG promotes truthful reporting through mutual information signals, which more effectively elicit the latent capabilities of the models. Table 2 further supports this, showing that each individual discriminator consistently improves in accuracy following policy updates under PEG.

**PEG Enables Coordination Among Heterogeneous Agents.** Despite substantial differences in architecture and baseline accuracy, Table 2 shows that all discriminators benefit from PEG. Remarkably, even the weakest model, OQwen-7B, which initially has the lowest accuracy among all agents, becomes the most accurate after participating in PEG. This suggests that PEG encourages cross-agent learning, where each model learns to align with truthful, high-confidence signals provided by its peers. Figure 3 reports the accuracy of each individual discriminator, as well as the majority vote, after applying PEG. The majority vote achieves a notable accuracy gain over any single model, validating the benefits of collaborative learning among heterogeneous agents.

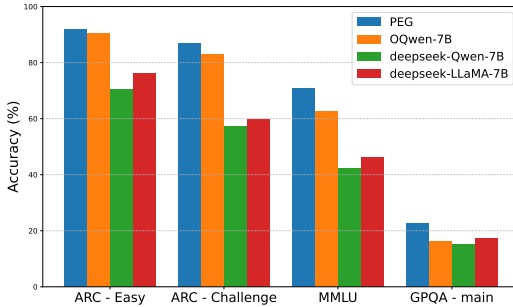

Figure 3: Accuracy comparison between original model outputs (D) and PEG majority vote answers.

Table 3: Accuracy (%) of individual discriminators before and after applying PEG in the 5-discriminator setting.

| 5 Discriminators | OQwen-7B | deepseek-Qwen-7B | deepseek-Llama2-7B | Gemma-7B | Mistral-7B |
| --- | --- | --- | --- | --- | --- |
| **Original** | 82.91 | 57.18 | 59.83 | 69.32 | 70.26 |
| **After PEG** | 86.84 | 84.27 | 83.76 | 84.27 | 82.74 |

**PEG with Varying Number of Discriminators.** We further evaluate PEG on the ARC-Challenge dataset with extended discriminator settings: 5-discriminators (adding Gemma-7B [53] and Mistral-7B [29]) and 7-discriminators (further adding Ai-Yi-9B [66] and OpenChat-7B [57]). The results consistently highlight the strong impact of PEG on both individual and collective performance. Notably, the weakest models with the lowest initial accuracy benefit the most from PEG. For example, in Tables 3 and 4, Qwen-7B and LLaMA2-7B gain over 20% improvement after applying PEG, whereas initially stronger models such as OQwen-7B and Ai-Yi-9B exhibit only marginal gains. Nevertheless, all models, regardless of their initial performance, converge toward coordinated outcomes after PEG. These results demonstrate that agents indeed learn from one another and achieve coordination through PEG. Moreover, when varying the number of discriminators, PEG

consistently outperforms both initial discriminator majority vote D and ER-D, especially with 3 and 5 discriminators, where it achieves over 10% improvement, as shown in Table 5.

Table 4: Accuracy (%) of individual discriminators before and after applying PEG in the 7-discriminator setting.

| 7 Discriminators | OQwen-7B | Qwen-7B | Llama2-7B | Gemma-7B | Mistral-7B | Ai-Yi-9B | OpenChat-7B |
|---|---|---|---|---|---|---|---|
| **Original** | 82.91 | 57.18 | 59.83 | 69.32 | 70.26 | 81.28 | 79.06 |
| **After PEG** | 83.85 | 75.30 | 73.08 | 77.61 | 79.06 | 81.62 | 82.99 |

Table 5: Overall accuracy (%) comparison of initial discriminator majority vote D, ER-D and PEG under 3-, 5-, and 7-discriminator settings. **Bold** indicates the best performance in each row.

| Setting | D | ER-D | PEG |
|---|---|---|---|
| 3 Discriminators | 70.68 | 77.52 | **87.01** |
| 5 Discriminators | 71.71 | 76.32 | **86.75** |
| 7 Discriminators | 76.50 | 81.54 | **81.97** |

Finally, we observe a slight decrease in accuracy when expanding to seven discriminators. This can be attributed to a mild violation of the conditional independence assumption (Assumption 4 in Appendix A.1), which ensures that the mutual information–based utility remains informative and non-degenerate. Intuitively, when discriminators are highly similar, their outputs become redundant, i.e., observing one provides little new information beyond the others. As a result, the mutual-information-based utility degenerates to zero, so truthfulness no longer maximizes the utility. Consequently, the learning dynamics may converge to a suboptimal equilibrium where discriminators are not incentive-compatible in (1), resulting in a reduction in accuracy. Overall, our results indicate that PEG remains robust with up to five discriminators; see, Table 5. As a practical guideline, we recommend using three to five heterogeneous LLMs, which provide a strong balance between performance, stability, and computational efficiency. Expanding beyond this should be done cautiously and only when sufficient diversity and independence among LLMs can be ensured.

## 5    Conclusions

In this paper, we propose a training-free, game-theoretic framework for aligning LLMs through a multi-agent peer elicitation game. Through mutual evaluations among agents, our peer elicitation game facilitates interactions between a generator and multiple discriminators in a way that provably incentivizes truthful behavior. We theoretically show that truthful reporting is a dominant strategy for each discriminator. Furthermore, using online mirror descent, each agent achieves sublinear regret, ensuring that its average performance approaches that of the best fixed truthful strategy. The agents' strategies also converge in the last iterate to a truthful Nash equilibrium. Empirically, our framework significantly improves factual accuracy across a range of benchmarks and performs competitively with much larger models, highlighting a practical direction for deploying lightweight models in resource-constrained environments.

There are several promising directions for future research based on PEG. One is to extend PEG to high-stakes settings such as medical decision support, scientific fact verification, and policy-relevant summarization, where truthful and consistent outputs are essential for safety and reliability. Another is to incorporate concepts from game theory and economics, such as reputation systems, repeated interactions, and budget-aware mechanisms, to further enhance alignment and robustness among LLM agents, particularly in open-ended or adversarial environments.

## Acknowledgments and Disclosure of Funding

We would like to thank the area chair and four anonymous referees for constructive suggestions that improve the paper. X. Dai was supported in part by NIH grant R01DK142026, a Merck Research Award, and a Hellman Fellowship Award.

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

# Appendix

Appendix A contains the proofs of the theoretical results presented in this paper. Specifically, Appendix A.1 presents the proof of Lemma 1, Appendix A.2 provides the proof of Theorem 1, and Appendix A.3 includes the proof of Theorem 2. Appendix B provides the pseudocode of the PEG algorithm. Appendix C offers additional experimental details related to Section 4.

## A  Proofs

### A.1  Proof of Lemma 1

Formally, we consider a binary-choice setting, where each task consists of two possible outcomes, denoted as $\{0, 1\}$. There are $n$ agents. Each agent $i$ is assigned $K$ binary-choice tasks, where for each task $k$, agent $i$ receives a private signal $c_i^k \in \{0, 1\}$. These private signals for all agents are drawn from a joint unknown prior distribution $U_{[n]}^k \in \Delta_{[n]}^2$, where $\Delta_{[n]}^2$ represents the set of all measurable distributions over $\{0, 1\}^n$. For the same task, the private signals of different agents are correlated. For different tasks, the private signals of the same agent independent. In our setting, discriminators do not incur effort to acquire their private signals.

For a multi-task peer evaluation mechanism, the agents do not know the specific realization of the prior distribution before receiving the private signals. After receiving the private signal $c_i^k$, each agent is required to report $r_i^k$, which may or may not reflect their true signal. The mechanism is designed to incentivize agents to truthfully report their signals.

**Assumption 2.** *[A Priori Similar Tasks] We assume that all tasks are drawn from a common unknown prior distribution $U_{[n]}$ such that $U_{[n]}^k = U_{[n]}$ for all tasks $k$.*

This assumption states that all tasks are fundamentally similar, implying that the signals across different tasks should follow the same joint distribution. Specifically, the signals $\mathbf{c}^k$ for each task $k$ are assumed to be i.i.d.. While traditional single-task peer evaluation typically assumes a homogeneous prior, this multi-task setting allows for heterogeneity in agents' beliefs as long as the tasks themselves are similar. In our setting, this assumption implies that the problems are similar in the sense that the ground truths follow the same joint distribution within each batch at every round. We enforce this by ensuring that the same set of subjects and the same set of problems are presented in each round.

**Definition 1** (Strategy). *Each agent $i$'s strategy for reporting is a mapping from her private signal $c_i^k$ to a distribution over possible reports $r_i^k$. Formally, a strategy can be represented as a function $S_i^k : \{0, 1\} \to \Delta(\{0, 1\})$, where $S_i^k(c_i^k)$ gives the probability distribution over the possible reports $r_i^k$ conditioned on receiving the private signal $c_i^k$.*

Every strategy $S_i^k$ corresponds to a $2 \times 2$ transition matrix where $S_i^k(c_i^k, r_i^k)$ is the probability that agent $i$ reports $r_i^k$ given that she receives private signal $c_i^k$. A strategy is *truthful* if the agent always reports $r_i^k = c_i^k$. Agent $i$ plays a *truthful strategy* if for every task $k$, $S_i^k$ is an identity matrix.

**Assumption 3** (Consistent Strategy). *Each agent $i$ plays the same strategy $S_i$ for all tasks.*

This assumption is reasonable because agents face structurally identical tasks drawn from the same distribution, with no task-specific information to adapt to.

**Assumption 4** (Conditional Independence). *We assume that agents' private signals $c_1, c_2, ...c_n$ are independent conditioning on ground truth. Since agents' strategies are independent, this also implies that agent's reports $\hat{c}_1, \hat{c}_2, ...\hat{c}_n$ are independent conditioning on ground truth.*

In our case, the discriminators (agents) are instantiated independently and process inputs separately, so their reports are naturally conditionally independent given the underlying truth.

**Definition 2** (Informative Peer). *Agent $i$ and agent $j$ are considered each other's informative peers if the determinant of the joint distribution matrix over their private signals $\mathbf{c}_i$ and $\mathbf{c}_j$ is non-zero, i.e., $\det(U_{\mathbf{c}_i, \mathbf{c}_j}) \neq 0$, where $U_{\mathbf{c}_i, \mathbf{c}_j}$ represents the joint prior distribution of the signals $\mathbf{c}_i$ and $\mathbf{c}_j$, and $U_{\mathbf{c}_i, \mathbf{c}_j}$ is expressed in its matrix form.*

Definition 2 captures whether two agents are "informative peers" by examining the structure of their shared information. If their private signals are sufficiently correlated (as captured by a non-zero

determinant of the joint distribution matrix), they can serve as reliable references for each other in PEG.

**Definition 3.** *Given two random variables $X, Y$ which have the same support $\mathcal{C}$, we define the determinant mutual information (DMI) between $X$ and $Y$ as*

$$DMI(X;Y) = |\det(\mathbf{U}_{X,Y})|.$$

**Lemma 2** (Strict Information Monotonicity). *For every two random variables $X, Y$ with the same support $\mathcal{C}$, when $X'$ is less informative than $X$, i.e., $X'$ is independent of $Y$ conditioning on $X$, it holds that*

$$\mathrm{DMI}(X';Y) \leq \mathrm{DMI}(X;Y).$$

*The inequality is strict when $\det(\mathbf{U}_{X,Y}) \neq 0$ and $\mathbf{U}_{X'|X}$ is not a permutation matrix.*

**Lemma 3.** *Let $\mathbf{M}_\ell^{ij}$ be the co-occurrence matrix formed by agents $i$ and $j$ over the set of tasks $\mathcal{T}_\ell$, and define the score $\det(\mathbf{M}_\ell^{ij})$. Then the expectation of this score is an unbiased estimator of the determinant mutual information between $\hat{X}_i$ and $\hat{X}_j$:*

$$\mathbb{E}_{\hat{X}_i, \hat{X}_j} \left[ \det(\mathbf{M}_\ell^{ij}) \right] = a_\ell \cdot \det(\mathbf{U}_{\hat{X}_i, \hat{X}_j}),$$

*where $a_\ell = \binom{|\mathcal{T}_\ell|}{|\mathcal{C}|} \cdot |\mathcal{C}|!$ and $\mathbf{U}_{\hat{X}_i, \hat{X}_j}$ is the co-occurrence matrix of the random variables $\hat{X}_i$ and $\hat{X}_j$.*

The proofs of Lemmas 2 and 3 follow the argument of Theorem 5.1 in [33]. Building on these results, we now present the proof of Lemma 1.

*Proof.* Let agent $i$ truthfully report $\hat{X}_i = X_i$, and let other agents report signals $\hat{X}_j$. Consider the interaction between agent $i$ and any peer agent $j$. The DMI between $X_i$ and $\hat{X}_j$ satisfies:

$$\mathrm{DMI}(\hat{X}_i; \hat{X}_j) \leq \mathrm{DMI}(X_i; \hat{X}_j),$$

with equality if and only if $\hat{X}_i$ is a permutation of $X_i$. In particular, if $\hat{X}_j = X_j$ (agent $j$ reports truthfully), then:

$$\mathrm{DMI}(\hat{X}_i; X_j) \leq \mathrm{DMI}(X_i; X_j),$$

and the inequality is strict if $\hat{X}_i$ is not a permutation of $X_i$, due to the strict information monotonicity of DMI. Therefore, the truthful strategy maximizes DMI against any truthfully reporting peer.

From Lemma 3, we know that the utility between agents $i$ and $j$, based on co-occurrence matrix $\mathbf{M}_\ell^{ij}$, satisfies:

$$\mathbb{E}[\det(\mathbf{M}_\ell^{ij})] = a_\ell \cdot \det(\mathbf{U}_{\hat{X}_i, \hat{X}_j}),$$

where $a_\ell$ is a constant depending on the number of tasks in round $\ell$, and $\mathbf{U}_{\hat{X}_i, \hat{X}_j}$ is the empirical joint distribution matrix. Therefore, the expected squared utility is proportional to:

$$\{\mathbb{E}[\det(\mathbf{M}_\ell^{ij})]\}^2 \propto \mathrm{DMI}^2(\hat{X}_i; \hat{X}_j),$$

and is maximized when $\hat{X}_i = X_i$, i.e., the agent reports truthfully. Since each agent's expected utility (aggregated over peers $j \neq i$) is a weighted sum of $\mathrm{DMI}^2(\hat{X}_i; \hat{X}_j)$, and each term in the sum is maximized by truthful reporting, it follows that truth-telling is a dominant strategy. Furthermore, when all agents report truthfully, the resulting strategy profile forms an equilibrium, and any deviation leads to a strictly lower expected utility unless the deviation is a permutation of the truth. $\qquad\square$

## A.2 Proof of Theorem 1

To maintain notational clarity and avoid redundancy, we use $t$ to replace the notation $t'$ used in the main text in Appendix A.2 and A.3.

**Definition 4** (Bregman divergence). *Let $\varphi : \Omega \to \mathbb{R}$ be a differentiable and $\mu$-strongly convex ($\mu > 0$) function with respect to a norm $\|\cdot\|$, that is, satisfy*

$$\varphi(x) \geq \varphi(y) + \langle \nabla\varphi(y), x - y \rangle + \frac{\mu}{2}\|x - y\|^2, \quad \forall x, y \in \Omega.$$

*The Bregman divergence centered in $y \in \Omega$ is the function $D_\varphi(x \parallel y)$ defined as*

$$D_\varphi(x \parallel y) := \varphi(x) - \varphi(y) - \langle \nabla\varphi(y), x - y \rangle.$$

**Definition 5.** *Let $f : \Omega \to \mathbb{R}$ be a convex function, and let $D_\varphi$ be a Bregman divergence. The proximal mapping (or proximal step) for a point $x_t$ with step size $\eta > 0$ is defined as:*

$$\text{Prox}_\varphi(\eta \nabla f(x_t), x_t) = \arg\min_{x \in \Omega} \{\eta \langle \nabla f(x_t), x \rangle + D_\varphi(x \| x_t)\}.$$

*The mirror descent algorithm is then given by the iterative update:*

$$x_{t+1} := \text{Prox}_\varphi(\eta \nabla f(x_t), x_t).$$

**Lemma 4.** *Let $\pi_\theta(y|x)$ be a policy and $R(y)$ be a utility function independent of $\theta$. Define the objective function:*

$$J(\theta) = \mathbb{E}_{y \sim \pi_\theta(\cdot|x)} [R(y)].$$

*Then, the gradient of $J(\theta)$ with respect to $\theta$ is:*

$$\nabla_\theta J(\theta) = \mathbb{E}_{y \sim \pi_\theta(\cdot|x)} [R(y) \nabla_\theta \log \pi_\theta(y|x)].$$

*Proof.*

$$\nabla_\theta J(\theta) = \nabla_\theta \int \pi_\theta(y|x) R(y) \, dy = \int \nabla_\theta \pi_\theta(y|x) R(y) \, dy$$

$$= \int \pi_\theta(y|x) \nabla_\theta \log \pi_\theta(y|x) \cdot R(y) \, dy$$

$$= \mathbb{E}_{y \sim \pi_\theta(\cdot|x)} [R(y) \nabla_\theta \log \pi_\theta(y|x)].$$

$\square$

**Corollary 3** (Gradient of utility Functions). *Using Lemma 4, the gradients of the utility functions with respect to the discriminator and generator parameters are given by: For each discriminator $D_i$:*

$$\nabla_{\theta_{D_i}} u_{D_i} = \mathbb{E}_{r_{ik} \sim \pi_{D_i}(\cdot|x_k, y_k)} \left[ \left( \sum_{j \neq i} \det(M_{ij}^1) \cdot \det(M_{ij}^2) \right) \nabla_{\theta_{D_i}} \log \pi_{D_i}(r_{ik}|x_k, y_k) \right].$$

*For the generator $G$:*

$$\nabla_{\theta_G} u_G = \mathbb{E}_{y_k \sim \pi_G(\cdot|x_k, v_k)} [\mathbf{1}(v_k = \hat{v}_k) \nabla_{\theta_G} \log \pi_G(y_k|x_k, v_k)].$$

**Lemma 5.** *When $\Omega = \triangle^n$ is the set of full-support distributions over $n$ objects and the $\varphi$ is set to the negative entropy function, which is 1-strongly convex with respect to the $\ell_1$ norm $\| \cdot \|_1$, the corresponding Bregman divergence is the Kullback-Leibler (KL) divergence.*

**Lemma 6.** *Let $x' = \text{Prox}_\varphi(g, x)$ be the proximal update. Then, for all $y \in \Omega$, we have:*

$$\langle -g, y - x' \rangle \leq -D_\varphi(y \| x') + D_\varphi(y \| x) - D_\varphi(x' \| x).$$

*By setting $y = x$, this three-point inequality simplifies to*

$$\langle -g, x - x' \rangle \leq -D_\varphi(x \| x') - D_\varphi(x' \| x).$$

*Proof.* The objective function of the proximal step problem is given by

$$h(z) := \langle g, z \rangle + D_\varphi(z \| x), \quad z \in \Omega.$$

The first-order optimality conditions applied to the solution $z = x'$ are therefore

$$
\begin{aligned}
-\nabla h(x') \in \mathcal{N}_\Omega(x') &\iff -g - \nabla \varphi(x') + \nabla \varphi(x) \in \mathcal{N}_\Omega(x') \\
&\iff \langle -g - \nabla \varphi(x') + \nabla \varphi(x), y - x' \rangle \leq 0 \quad \forall y \in \Omega \\
&\iff \langle -g, y - x' \rangle \leq \langle \nabla \varphi(x') - \nabla \varphi(x), y - x' \rangle \quad \forall y \in \Omega.
\end{aligned}
$$

The statement now follows from using the identity

$$\langle \nabla \varphi(x') - \nabla \varphi(x), y - x' \rangle = -D_\varphi(y \| x') + D_\varphi(y \| x) - D_\varphi(x' \| x),$$

which can be checked directly from the definition of Bregman divergence. $\square$

**Lemma 7.** *Let* $f : \Omega \to \mathbb{R}$ *be convex. Each step of the mirror descent algorithm satisfies*

$$f(x_t) \leq f(y) + \langle \nabla f(x_t), x_t - x_{t+1} \rangle - \frac{1}{\eta_t} D_\varphi(y \parallel x_{t+1}) + \frac{1}{\eta_t} D_\varphi(y \parallel x_t) - \frac{1}{\eta_t} D_\varphi(x_{t+1} \parallel x_t).$$

*Proof.* Using the linear lower bound property of convex functions, we can write

$$f(x_t) \leq f(y) - \langle \nabla f(x_t), y - x_t \rangle = f(y) + \langle \nabla f(x_t), x_t - x_{t+1} \rangle - \langle \nabla f(x_t), y - x_{t+1} \rangle.$$

On the other hand, from Lemma 6 applied to the mirror descent step (that is, for the choices $g = \eta_t \nabla f(x_t), x' = x_{t+1}, x = x_t$), we have

$$-\eta_t \langle \nabla f(x_t), y - x_{t+1} \rangle \leq -D_\varphi(y \| x_{t+1}) + D_\varphi(y \| x_t) - D_\varphi(x_{t+1} \| x_t).$$

Hence, dividing by $\eta_t$ and plugging into the previous inequality, we obtain the statement.

$\square$

**Lemma 8.** *Let* $\| \cdot \|$ *be the norm with respect to which the DGF* $\varphi$ *is 1-strongly convex, and* $\| \cdot \|_*$ *be the dual norm. If all functions* $f_t : \Omega \to \mathbb{R}$ *are convex, the regret of online mirror descent is bounded by*

$$R_T := \sum_{t=1}^{T} f_t(x_t) - \min_{x \in \Omega} \sum_{t=1}^{T} f_t(x) \leq \frac{1}{\eta} D_\varphi(x \| x_1) + \frac{\eta}{2} \sum_{t=1}^{T} \| \nabla f_t(x_t) \|^2.$$

*In particular, assuming that all dual gradient norms are upper bounded by G, and setting*

$$\eta := \sqrt{\frac{2 D_\varphi(x \| x_1)}{M^2 \cdot 2^i}} \quad for \ 2^i \leq T < 2^{i+1}.$$

*we find*

$$R_T \leq M \sqrt{2 D_\varphi(x \| x_1) \cdot T} \frac{\sqrt{2}}{\sqrt{2} - 1}.$$

*Proof.* Since functions $f_t$ are convex, we can use Lemma 7:

$$f_t(x_t) \leq f_t(x) + \langle \nabla f_t(x_t), x_t - x_{t+1} \rangle - \frac{1}{\eta_t} D_\varphi(x \| x_{t+1}) + \frac{1}{\eta_t} D_\varphi(x \| x_t) - \frac{1}{\eta_t} D_\varphi(x_{t+1} \| x_t) \quad \forall x \in \Omega.$$

Using the Cauchy-Schwarz inequality, we can bound the right-hand side by

$$f_t(x_t) \leq f_t(x) + \| \nabla f_t(x_t) \|_* \cdot \| x_t - x_{t+1} \| - \frac{1}{\eta_t} D_\varphi(x \| x_{t+1}) + \frac{1}{\eta_t} D_\varphi(x \| x_t) - \frac{1}{\eta_t} D_\varphi(x_{t+1} \| x_t).$$

Using Young's inequality, as well as the 1-strong convexity of the KL divergence, which implies

$$\frac{1}{\eta_t} D_\varphi(x_{t+1} \| x_t) \geq \frac{1}{2\eta_t} \| x_{t+1} - x_t \|^2,$$

$$f_t(x_t) \leq f_t(x) + \frac{\eta_t}{2} \| \nabla f_t(x_t) \|^2 + \frac{1}{2\eta_t} \| x_t - x_{t+1} \|^2 - \frac{1}{\eta_t} D_\varphi(x \| x_{t+1}) + \frac{1}{\eta_t} D_\varphi(x \| x_t) - \frac{1}{2\eta_t} \| x_{t+1} - x_t \|^2.$$

Rearranging terms, we obtain:

$$f_t(x_t) \leq f_t(x) + \frac{\eta_t}{2} \| \nabla f_t(x_t) \|^2 - \frac{1}{\eta_t} D_\varphi(x \| x_{t+1}) + \frac{1}{\eta_t} D_\varphi(x \| x_t).$$

Summing over $t = 1, \ldots, T$:

$$\sum_{t=1}^{T} (f_t(x_t) - f_t(x)) \leq \frac{1}{2} \sum_{t=1}^{T} \eta_t \| \nabla f_t(x_t) \|_2^2 \| + \sum_{t=1}^{T} \frac{1}{\eta_t} (D_\varphi(x \| x_t) - D_\varphi(x \| x_{t+1})).$$

The regret incurred by the algorithm is upper bound by the regret incurred in each of the intervals $2^i \leq T < 2^{i+1}$. Suppose the algorithm has been run until time $2^i \leq T < 2^{i+1}$. Hence, the regret is upper bounded by

$$\text{Reg}_T \leq \left( M\sqrt{\frac{D_\varphi(x\|x_1)}{2}} \sum_{i=0}^{I} (\sqrt{2})^i \right) + \left( \sum_{i=0}^{I-1} \frac{\sqrt{D_\varphi(x\|x_1)}}{2M^2 2^i} M^2 2^i \right) + \left( \sqrt{\frac{2D_\varphi(x\|x_1)}{2M^2 2^I}} \right) M^2 (T - 2^I)$$

$$= M\sqrt{\frac{D_\varphi(x\|x_1)}{2}} \left( \sum_{i=0}^{I} (\sqrt{2})^i + \sum_{i=0}^{I-1} (\sqrt{2})^i \right) + \sqrt{\frac{D_\varphi(x\|x_1)}{2M^2 2^I}} M^2 (T - 2^I)$$

In particular, since $T < 2^{I+1}$,

$$\text{Reg}_T \leq M\sqrt{\frac{D_\varphi(x\|x_1)}{2}} \left( \sum_{i=0}^{I} (\sqrt{2})^i + \sum_{i=0}^{I-1} (\sqrt{2})^i \right) + \sqrt{\frac{D_\varphi(x\|x_1)}{2M^2 2^I}} M^2 2^I$$

$$= M\sqrt{\frac{D_\varphi(x\|x_1)}{2}} \left( \sum_{i=0}^{I} (\sqrt{2})^i + \sum_{i=0}^{I} (\sqrt{2})^i \right)$$

$$= M\sqrt{2D_\varphi(x\|x_1)} \sum_{i=0}^{I} (\sqrt{2})^i$$

$$= M\sqrt{2D_\varphi(x\|x_1)} \cdot \frac{(\sqrt{2})^{I+1} - 1}{\sqrt{2} - 1}$$

$$\leq M\sqrt{2D_\varphi(x\|x_1)T} \cdot \cdot \frac{\sqrt{2}}{\sqrt{2} - 1}.$$

$\square$

**Lemma 9.** *Suppose the utility is defined over a set of $K$ policies, and each policy is updated independently via a no-regret algorithm with individual regret bounded by $O(\sqrt{T})$. Then, the total regret of the system with respect to the best fixed policy in hindsight is bounded by $O(\sqrt{KT})$. Therefore, when both the generator and each discriminator maintain and update $K$ independent policies, their regret bounds will incur an additional $\sqrt{K}$ factor, yielding an overall regret of $O(\sqrt{KT})$ per agent.*

This result that the regret grows as $O(\sqrt{KT})$ when updating and aggregating over $K$ policies is a classical result in online learning and multi-armed bandits, as established in [5, 51]. Therefore, by combining Lemmas 8 and 9, we complete the proof of Theorem 1.

### A.3 Proof of Theorem 2

**Lemma 10.** *If the utility function is concave and differentiable, its gradient is monotone.*

*Proof.* By concavity, the first-order conditions for $u$ at $x$ and $y$ give:

$$u(y) \leq u(x) + \nabla u(x)^\top (y - x),$$
$$u(x) \leq u(y) + \nabla u(y)^\top (x - y).$$

Adding these inequalities:

$$u(y) + u(x) \leq u(x) + u(y) + \nabla u(x)^\top (y - x) + \nabla u(y)^\top (x - y).$$

Simplifying:

$$(\nabla u(x) - \nabla u(y))^\top (x - y) \leq 0.$$

Thus, $\nabla u$ is monotone:

$$(\nabla u(x) - \nabla u(y))^\top (x - y) \leq 0 \quad \forall x, y \in \mathcal{X}.$$

$\square$

**Lemma 11.** *Let $\pi_{i,t+1}$ be the update given by mirror descent:*

$$\pi_{i,t+1} = \arg\min_{\pi_i \in \Pi_i} \left\{ \eta_t \langle -\nabla u_i(\pi_t), \pi_i \rangle + D_i(\pi_i, \pi_{i,t}) \right\},$$

*where $D_i(\cdot, \cdot)$ is the Bregman divergence generated by a $\mu$-strongly convex function $h_i$. Then for any $\pi_i^* \in \Pi_i$, we have*

$$D_i(\pi_i^*, \pi_{i,t+1}) \le D_i(\pi_i^*, \pi_{i,t}) - \eta_t \langle \nabla V_i(\pi_t), \pi_{i,t} - \pi_i^* \rangle + \frac{L_i \eta_t^2}{2},$$

*where $L_i > 0$ is a constant that bounds $\|\nabla V_i(\pi_t)\|^2$ due to compactness of $\Pi_i$.*

*Proof.* Let $x = \pi_{i,t}$, $x^+ = \pi_{i,t+1}$, $z = \pi_i^*$, and $g_i = \nabla u_i(\pi_t)$. The mirror descent update is:

$$x^+ = \arg\min_{\pi_i \in \Pi_i} \left\{ -\eta_t \langle g_i, \pi_i \rangle + D_i(\pi_i, x) \right\}.$$

By the first-order optimality condition for convex minimization over $\Pi_i$, we have:

$$\langle -\eta_t g_i + \nabla h_i(x^+) - \nabla h_i(x), z - x^+ \rangle \ge 0,$$

which gives:

$$\eta_t \langle g_i, z - x^+ \rangle \le \langle \nabla h_i(x^+) - \nabla h_i(x), z - x^+ \rangle.$$

Using the three-point identity for Bregman divergence:

$$\langle \nabla h_i(x^+) - \nabla h_i(x), z - x \rangle = D_i(z, x) - D_i(z, x^+) - D_i(x^+, x),$$

we substitute and rearrange:

$$D_i(z, x^+) \le D_i(z, x) - \eta_t \langle g_i, z - x \rangle + \eta_t \langle g_i, x^+ - x \rangle - D_i(x^+, x).$$

Now apply Young's inequality:

$$\eta_t \langle g_i, x^+ - x \rangle \le \frac{1}{2\mu} \|x^+ - x\|^2 + \frac{\mu \eta_t^2}{2} \|g_i\|^2.$$

By strong convexity, $D_i(x^+, x) \ge \frac{\mu}{2} \|x^+ - x\|^2$, so:

$$\eta_t \langle g_i, x^+ - x \rangle \le D_i(x^+, x) + \frac{\eta_t^2 \|g_i\|^2}{2\mu}.$$

Substituting back, the $D_i(x^+, x)$ term cancels, and we obtain:

$$D_i(z, x^+) \le D_i(z, x) - \eta_t \langle g_i, z - x \rangle + \frac{\eta_t^2 \|g_i\|^2}{2\mu}.$$

Let $L_i := \frac{1}{\mu} \sup_{\pi \in \mathcal{K}} \|\nabla u_i(\pi)\|^2$, which is finite due to compactness of $\Pi_i$. Therefore:

$$D_i(\pi_i^*, \pi_{i,t+1}) \le D_i(\pi_i^*, \pi_{i,t}) - \eta_t \langle \nabla u_i(\pi_t), \pi_i^* - \pi_{i,t} \rangle + \frac{L_i \eta_t^2}{2}.$$

$\square$

Next we can move to the proof of Theorem 2.

*Proof.* We define an energy function that tracks the distance to the Nash equilibrium using Bregman divergences. For each player $i \in \mathcal{N}$, let $D_i$ denote the associated Bregman divergence:

$$D_i(\pi_i^*, \pi_{i,t}) := h_i(\pi_i^*) - h_i(\pi_{i,t}) - \langle \nabla h_i(\pi_{i,t}), \pi_i^* - \pi_{i,t} \rangle.$$

We define the total energy function as

$$E_t := \sum_{i \in \mathcal{N}} D_i(\pi_i^*, \pi_{i,t}).$$

Each player updates their strategy via mirror descent:

$$\pi_{i,t+1} = \arg\min_{\pi_i \in \Pi_i} \left\{ \eta_t \langle -\nabla u_i(\pi_t), \pi_i \rangle + D_i(\pi_i, \pi_{i,t}) \right\}.$$

With Lemma 11, we have:

$$D_i(\pi_i^*, \pi_{i,t+1}) \leq D_i(\pi_i^*, \pi_{i,t}) - \eta_t \langle \nabla u_i(\pi_t), \pi_{i,t} - \pi_i^* \rangle + \frac{L_i \eta_t^2}{2}.$$

Summing over all players, we obtain:

$$E_{t+1} \leq E_t - \eta_t \langle \nabla u(\pi_t), \pi_t - \pi^* \rangle + C\eta_t^2,$$

for some constant $C > 0$, where $\nabla V(\pi_t) = (\nabla u_1(\pi_t), \ldots, \nabla u_N(\pi_t))$.

Because the utility map $\nabla u$ is strictly monotone, there exists $\mu > 0$ such that

$$\langle \nabla u(\pi_t), \pi_t - \pi^* \rangle \geq \mu \|\pi_t - \pi^*\|^2.$$

Substituting into the previous inequality yields:

$$E_{t+1} \leq E_t - \mu \eta_t \|\pi_t - \pi^*\|^2 + C\eta_t^2.$$

This recursion is in the form required by the Robbins–Siegmund Lemma:

$$E_{t+1} \leq E_t - a_t + b_t,$$

where $a_t = \mu \eta_t \|\pi_t - \pi^*\|^2 \geq 0$ and $b_t = C\eta_t^2$, with $\sum_{t=1}^{\infty} b_t < \infty$ since $\eta_t = O(1/t^p)$ with $2p > 1$. Therefore, the Robbins–Siegmund Lemma implies that $E_t$ converges almost surely to a finite random variable $E_\infty$, and $\sum_t \eta_t \|\pi_t - \pi^*\|^2 < \infty$. Since $\sum_t \eta_t^2 < \infty$ and $\sum_t \eta_t = \infty$, $\|\pi_t - \pi^*\| \to 0$ almost surely. $\qquad\square$

## B  PEG Algorithm

### B.1  Algorithm

---

**Algorithm 1** Two-Phase PEG Algorithm with Task Batches

---

**Require:** Dataset of questions $\{x_k\}$ (organized into batches of size 8 in the experiments); initial parameters $\theta_G$ for generator; initial parameters $\{\theta_{D_i}\}_{i=1}^n$ for discriminators; learning rate $\eta$.

1: **repeat**
2:     **for** each batch **do**
3:         Generator samples answers: $y_k \sim \pi_G(\cdot | x_k, v_k)$.
4:         **(Step 1) Discriminator Updates:**
5:         **for** each discriminators $D_i$ **do**
6:             *(a) Discriminators provide judgments:* $r_{ik} \sim \pi_{D_i}(\cdot | x_k, y_k)$.
7:             *(b) Approximate gradients for each $D_i$ using the log-derivative trick:*

$$\nabla_{\theta_{D_i}} u_{D_i} \approx \frac{1}{N} \sum_{\text{batches}} \left[ \sum_{j \neq i} \det(M_{ij}^1) \cdot \det(M_{ij}^2) \right] \nabla_{\theta_{D_i}} \log \pi_{D_i}(r_{ik} | x_k, y_k).$$

8:             *(c) OMD update for discriminator policies:* $\pi_{D_i}^{(t+1)} \propto \pi_{D_i}^{(t)} \exp\left(\eta \, \nabla_{\theta_{D_i}} u_{D_i}\right)$.
9:         **end for**
10:         **(Step 2) Generator Update:**
11:         *(a) Compute majority vote $\hat{v}_k$ for each question $x_k$ in the batch.*
12:         *(b) Approximate gradient for the generator:*

$$\nabla_{\theta_G} u_G \approx \frac{1}{N} \sum_{\text{batches}} \mathbf{1}(v_k = \hat{v}_k) \nabla_{\theta_G} \log \pi_G(y_k | x_k, v_k).$$

13:         *(c) PEG update for generator policy:* $\pi_G^{(t+1)} \propto \pi_G^{(t)} \exp\left(\eta \, \nabla_{\theta_G} u_G\right)$.
14:     **end for**
15: **until** *all policy have been updated*
16: **Output:** final parameters $\theta_G$ and $\{\theta_{D_i}\}_{i=1}^n$.

---

## B.2 Illustrative example

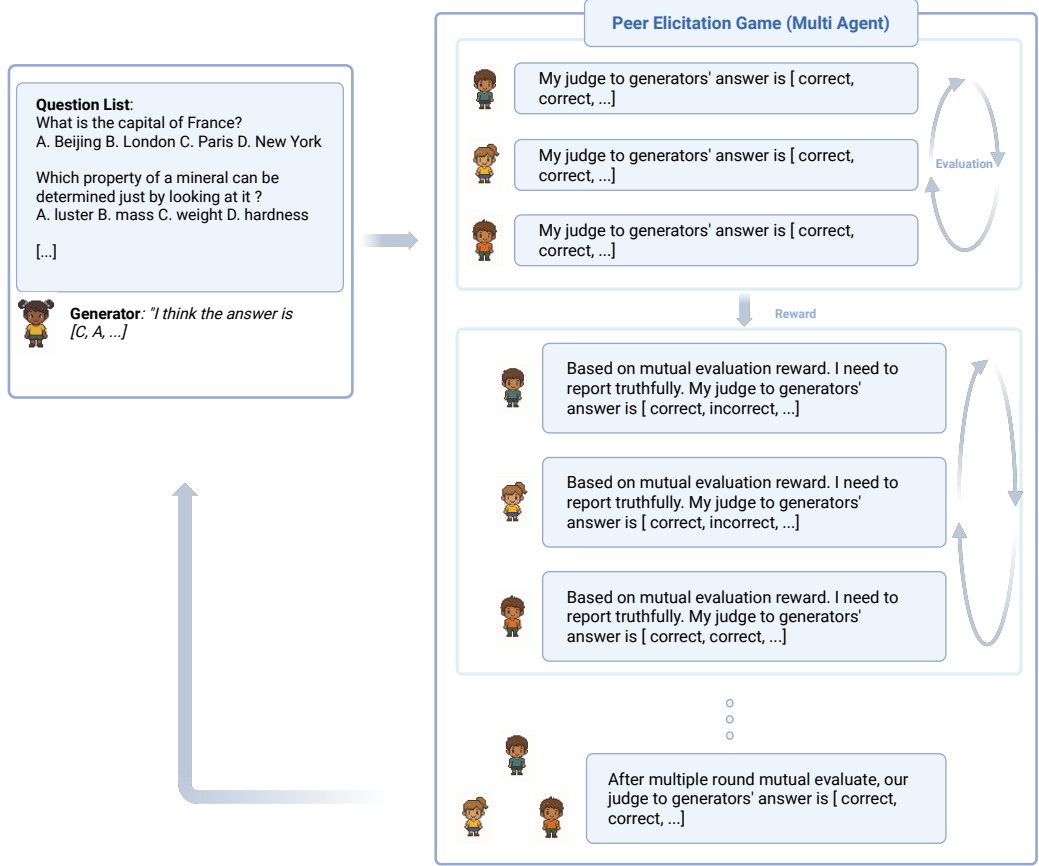

Figure 4: An illustrative example of PEG's peer evaluation process: (1) The generator answers a list of questions. (2) Discriminators evaluate these answers, with some providing untruthful reports. (3) Determinant-based utilities penalize non-truthful discriminators, incentivizing them to align their future reports with the ground truth.

## C Experiments Details

### C.1 Prompt

**Generator Prompt**

The following are multiple choice questions (with answers) about Geology. Question: Which property of a mineral can be determined just by looking at it?

    A) luster

    B) mass

    C) weight

    D) hardness

Answer:

## C.2 Datasets

Datasets from different domains are used in the experiments. Details are summarized below.

- **ARC** dataset [14] evaluates scientific reasoning, includes both the 'easy' and 'challenge' sets. It consists of 7787 science questions, all nondiagrams, multiple choice (typically 4-way multiple choice). The experiment applies a zero-shot setting for ARC, assessing how well the model navigates science-related questions requiring logical reasoning rather than memorized knowledge.

- **MMLU** [21] is a benchmark with questions across humanities, STEM, and social sciences. It requires the model to demonstrate broad general knowledge. For MMLU, we evaluate our models following the format described in setting [21].

- **GPQA** [48] is a challenging dataset designed to evaluate LLM capabilities and scalable oversight mechanisms. It consists of 448 multiple choice questions that cover biology, physics, and chemistry. These questions are intentionally designed to be high-quality and extremely difficult. The experiment will apply the zero-shot setting for GPQA.

## C.3 Computational Resources

Our experiments utilized the following hardware configurations:

- **Initial Policy Extraction:**

  - **GPU:** A single NVIDIA A100 GPU with 200 GB of memory.

  - **CPU:** AMD EPYC 7542 or 9654 processors.

  - **Throughput:** Approximately 2.2 iterations per second (it/s) for policy generation per question.

  - **Estimated Runtime:** Varies by dataset size, typically requiring 10 minutes for 1000 questions.

- **PEG Algorithm for Policy Update:**

  - **GPU:** NVIDIA RTX 2080 Ti.

  - **Runtime:** Significantly faster, all datasets under 20 seconds.

## C.4 The impact of batch size

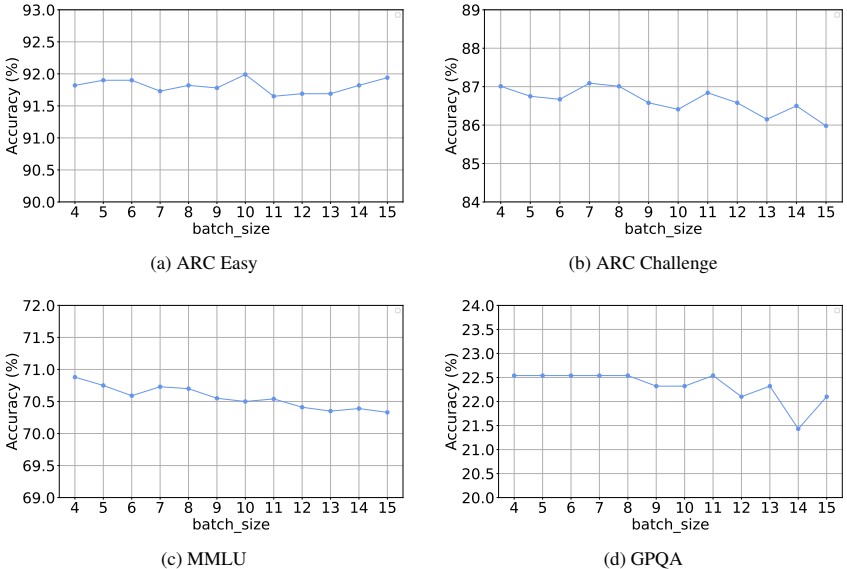

(a) ARC Easy

(b) ARC Challenge

(c) MMLU

(d) GPQA

Figure 5: Different Batch Size Effect on Majority Vote Accuracy

We conducted experiments on four benchmark datasets by varying the discriminator update batch size from 4 to 15 (with 4 being the minimum required to form a valid batch in this setup) to evaluate its impact on majority vote accuracy. As shown in Figure 5, the accuracy remains relatively stable across batch sizes, though slight fluctuations can be observed depending on the dataset. For ARC-Easy, the accuracy is consistently high and robust to batch size, suggesting stable discriminator learning on simpler, more homogeneous questions. In contrast, the other datasets display more variability and a slight decreasing trend as batch size increases. This performance drop may due to the greater diversity among questions within larger batches, making it harder for the discriminators to agree and learn from a consistent utility signal. These findings align with Assumption 2, which presumes that tasks within a batch are drawn from a common underlying distribution. Overall, the PEG algorithm demonstrates robustness to batch size variation when the assumption of task similarity holds.

## C.5  Sample Outputs

| Question | Generator's Answer | Discriminator #1 (OQwen-7B) | Discriminator #2 (deepseek-Qwen-7B) | Discriminator #3 (deepseek-LLaMA-7B) | Majority Vote (D) |
|---|---|---|---|---|---|
| To determine how closely related organisms are, scientists consider all of the following except | analogous structures | Incorrect | Incorrect | Correct | ❌ |
| Which is an example of a learned behavior? | A mouse runs from a coyote | Correct | Correct | Correct | ❌ |
| Suppose 20 g of liquid hydrogen peroxide is heated so it completely breaks down into liquid water and oxygen gas. Which best describes the total mass of the water and oxygen that was produced? | 20 g because no matter is added or removed | Correct | Correct | Incorrect | ✅ |
| Air has no color and cannot be seen, yet it takes up space. What could be done to show that air takes up space? | The Sun influences the formation of wavesblow up a beach ball or balloon | Correct | Correct | Correct | ✅ |
| Which geologic process most likely caused the formation of the Mount St. Helens Volcano? | converging boundaries | Correct | Correct | Correct | ✅ |
| Which example shows a relationship between a living thing and a nonliving thing? | A tree removes a gas from the air. | Incorrect | Incorrect | Incorrect | ❌ |
| Images from the Voyager and the Galileo spacecraft provide evidence Europa has a liquid ocean under a surface of ice that results in part from distinctive, surface-cracking patterns produced by which events? | tectonic movements | Incorrect | Correct | Incorrect | ✅ |
| All natural resources on Earth are either renewable or nonrenewable. Whether a resource is renewable or nonrenewable depends on how fast or slow the resource is replaced. If the resource is used faster than it is replaced, then the resource will, in time, disappear. Which activity shows the use of a nonrenewable natural resource? | A construction crew builds an iron bridge. | Correct | Correct | Correct | ✅ |

(a) Initial Discriminator Responses

| Question | Generator's Answer | Discriminator #1 (OQwen-7B) | Discriminator #2 (deepseek-Qwen-7B) | Discriminator #3 (deepseek-LLaMA-7B) | Majority Vote (PEG) |
|---|---|---|---|---|---|
| To determine how closely related organisms are, scientists consider all of the following except | analogous structures | Correct | Correct | Incorrect | ✅ |
| Which is an example of a learned behavior? | A mouse runs from a coyote | Correct | Correct | Incorrect | ❌ |
| Suppose 20 g of liquid hydrogen peroxide is heated so it completely breaks down into liquid water and oxygen gas. Which best describes the total mass of the water and oxygen that was produced? | 20 g because no matter is added or removed | Correct | Correct | Incorrect | ✅ |
| Air has no color and cannot be seen, yet it takes up space. What could be done to show that air takes up space? | The Sun influences the formation of wavesblow up a beach ball or balloon | Correct | Correct | Correct | ✅ |
| Which geologic process most likely caused the formation of the Mount St. Helens Volcano? | converging boundaries | Correct | Correct | Correct | ✅ |
| Which example shows a relationship between a living thing and a nonliving thing? | A tree removes a gas from the air. | Correct | Correct | Correct | ✅ |
| Images from the Voyager and the Galileo spacecraft provide evidence Europa has a liquid ocean under a surface of ice that results in part from distinctive, surface-cracking patterns produced by which events? | tectonic movements | Correct | Correct | Correct | ❌ |
| All natural resources on Earth are either renewable or nonrenewable. Whether a resource is renewable or nonrenewable depends on how fast or slow the resource is replaced. If the resource is used faster than it is replaced, then the resource will, in time, disappear. Which activity shows the use of a nonrenewable natural resource? | A construction crew builds an iron bridge. | Correct | Correct | Correct | ✅ |

(b) Updated Discriminator Responses via PEG

Figure 6: A batch example from the ARC-Challenge dataset showing discriminator responses before and after PEG-based policy updates. The top table illustrates the initial disagreement among discriminators, while the bottom table demonstrates improved convergence following utility-based updates. Red text indicates that the generator produced an incorrect answer. Red text indicates that the generator produced a incorrect answer. A green arrow highlights cases where the majority vote correctly judged the validity of the answer.

Figure6 shows a batch from the ARC-Challenge dataset that illustrates the discriminator responses before and after PEG-based policy updates. In the top table 6a, we observe noticeable disagreement among the three discriminators, reflecting their differing judgment capabilities across questions. For instance, Discriminator #3 (deepseek-LLaMA-7B) correctly identifies a case that the other two fail to judge, while in other questions it falls behind the others. After applying PEG 6b, the discriminators show improved agreement. In most cases, they converge to the correct judgment, leading to a higher overall decision accuracy.

## C.6  The impact of number of iterations

We conducted experiments on four benchmark datasets with {10, 20, 30, 40, 50} iterations using a fixed learning rate of 0.1 and a batch size of 8. From Table 6, we observe that PEG's performance is stable and does not require many iterations to achieve strong performance, with only minimal changes in accuracy and a few questions. We attribute this result to the fact that our updates operate directly in the output policy space rather than modifying model parameters, allowing for faster convergence within a few iterations.

Table 6: Accuracy (%) of PEG with different numbers of iterations across four benchmark datasets.

| Iterations | 10 | 20 | 30 | 40 | 50 |
|---|---|---|---|---|---|
| ARC-Challenge | 87.01 | 87.01 | 87.01 | 87.01 | 87.01 |
| ARC-Easy | 91.78 | 91.82 | 91.78 | 91.78 | 91.78 |
| MMLU | 70.78 | 70.81 | 70.81 | 70.79 | 70.81 |
| GPQA | 22.54 | 22.54 | 22.54 | 22.54 | 22.54 |

## C.7  The impact of learning rate

We conducted experiments on four benchmark datasets using a fixed number of 10 iterations with difference choices of learning rates in Table 7. Intuitively, smaller learning rates (e.g., $\leq 0.1$) yield stable performance without compromising convergence speed. In contrast, larger learning rates degrade performance, likely due to the fact that the updates are applied directly to the output distribution, which lies within a bounded space $[0, 1]$. As a result, overly aggressive updates may lead to oscillation or failure to converge.

Table 7: Accuracy (%) of PEG with different learning rates across four benchmark datasets.

| Learning Rate | 0.01 | 0.05 | 0.1 | 0.15 | 0.2 |
|---|---|---|---|---|---|
| ARC-Challenge | 87.01 | 87.01 | 87.01 | 86.50 | 80.68 |
| ARC-Easy | 91.82 | 91.82 | 91.82 | 91.65 | 88.44 |
| MMLU | 70.89 | 70.88 | 70.73 | 68.51 | 61.06 |
| GPQA | 22.54 | 22.54 | 22.54 | 22.32 | 19.42 |

## C.8  Impact of degree of initial disagreements

We conducted further analysis on the effectiveness of our method by categorizing problems in each dataset into different levels of difficulty based on the initial level of disagreement among the discriminators. We report the accuracy for each method across these groups on both ARC-Challenge and MMLU in Table 8. Our analysis reveals three key findings. First, PEG consistently improves accuracy across all levels of initial disagreement, suggesting that it enables agents to explore diverse judgments and learn from each other to reach better consensus. Second, a counter-intuitive finding is that cases with initial disagreement among discriminators actually result in higher accuracy compared to those with full agreement. We believe this phenomenon occurs because agreement alone does not ensure correctness: all discriminators may still converge on an incorrect answer. In contrast, disagreement introduces diversity, increasing the likelihood that at least one agent is correct, and serves as a useful signal to trigger further exploration. Third, PEG achieves the highest accuracy

across all levels compared to other baselines methods, validating its effectiveness under different settings.

Table 8: Accuracy (%) under different initial agreement settings for ARC-Challenge and MMLU datasets. **Bold** indicates the best performance in each row.

| Dataset | Setting | G | MI | ER-D | PEG |
|---|---|---|---|---|---|
| ARC-Challenge | Agreement | 64.31 | 73.85 | 72.79 | **84.10** |
| | Disagreement | 76.89 | 81.96 | 82.41 | **87.94** |
| MMLU | Agreement | 50.54 | 60.11 | 60.17 | **66.93** |
| | Disagreement | 57.96 | 65.19 | 65.02 | **72.20** |

# D  Potential Social Impact

Our proposed peer elicitation game contributes to the broader goal of building trustworthy and accountable language models by explicitly incentivizing truthful behavior through multi-agent interaction. By avoiding reliance on supervised fine-tuning and instead leveraging incentive-compatible mechanisms, the method has the potential to enhance factual reliability in resource-constrained or safety-critical settings such as education, healthcare information retrieval, and scientific communication. However, as with any mechanism involving agent-based interactions, care must be taken to ensure transparency in deployment and to prevent gaming of the system by adversarial agents.

