# OpenReview forum: "Incentivizing Truthful Language Models via Peer Elicitation Games"
_NeurIPS.cc/2025/Conference — NeurIPS 2025 poster_

### Official Review · Reviewer_6fgB · 2025-06-25

**Clarity:** 3
**Significance:** 3
**Originality:** 3
**Rating:** 4
**Confidence:** 3

**Summary:**

The paper introduces **Peer Elicitation Games (PEG)**, a training-free multi-agent framework in which a generator LLM answers questions and multiple discriminator LLMs evaluate the responses using a determinant-based peer-prediction reward. The authors theoretically prove that truthful reporting is a dominant strategy, with sub-linear regret and last-iterate convergence. PEG yields over 10% accuracy improvement compared to existing methods only using 4 open-source 7B models.

**Questions:**

1. Can you report PEG accuracy for varying discriminators and note any saturation or instability?
2. How does PEG accuracy change when each batch contains a mix of ARC-Challenge and MMLU items compared with single-domain batches?

**Ethical Concerns:**

["NO or VERY MINOR ethics concerns only"]

**Final Justification:**

The authors addressed the main concerns of guarantees rely on binary labels and conditionally independent signals, and supplemented ablation test on number and quality of discriminators. Thus, I increased the quality score by 1. While for the originality score, I mistakenly chose a low score of 2, so in the revision, I would like to correct it to 3.

**Limitations:**

Yes.

**Paper Formatting Concerns:**

The paper is well-formatted,

**Quality:**

4

**Strengths And Weaknesses:**

#### **Strengths**
1. Paper provides formal, rigorous proofs establishing dominant truthfulness, sub-linear regret, and last-iterate convergence guarantees throughout.
2. Truthful answers are elicited via pure black-box LLM calls, requiring no fine-tuning, labels, or RLHF, hence lightweight.
3. Demonstrates accuracy gains on ARC, MMLU, GPQA using only four open 7B models.

#### **Weaknesses**
1. Guarantees rely on binary labels and conditionally independent signals, limiting confidence in complex real-world language settings.
2. Experiments cover only multiple-choice QA, lacking tests on open-ended, factual, or safety tasks, limiting generality.
3. No ablations explore adversarial, low-quality, or fewer discriminators, leaving robustness, scalability, and heterogeneity effects unverified.

---

> ### Author Rebuttal · Authors · 2025-07-30
>
> We thank the reviewer for the insightful and supportive comments. Below, we do our best to address the reviewer's questions adequately such that we could receive a better score.
>
> **W1: Theoretical assumptions on binary labels and conditionally independent signals**
>
> Thanks for pointing out the validity of these assumptions in real-world language settings. We offer the following justifications to clarify our framework’s effectiveness:
>
> First, the generator is not restricted to a specific task type. PEG is compatible with open-ended tasks, as long as the generator can provide a set of candidate answers for evaluation. This makes the framework broadly applicable beyond multiple-choice QA.
>
> Second, our theoretical discussion adopts binary feedback for clarity, as the discriminators provide binary judgments in our setting. However, the DMI mechanism is originally designed for multi-choice settings and still guarantees dominant truthfulness under standard conditions. Specifically, the key requirement is that the batch size must be at least $2C$, where $C$ denotes the number of choices per question.
>
> Third, the conditional independence assumption is used in the theoretical analysis to ensure that the mutual information–based reward does not degenerate to $0$. Intuitively, if agents' signals are highly correlated, then observing others provides little to no additional information, and the utility (measured via a form of mutual information) collapses toward zero. In practice, since we use heterogeneous discriminators, their outputs are diverse and rarely reach consensus in the beginning. This natural diversity prevents the reward signal from collapsing and keeps the system informative.
>
> **W2: Clarification on task generality beyond multiple-choice QA**
>
> Thank you for the helpful suggestion. While our current experiments focus on multiple-choice QA tasks, PEG is not inherently limited to this setting. For any natural language task, we can use top-$k$ or other sampling methods to generate a candidate set, which PEG can then evaluate and refine. In this sense, the multiple-choice format is a practical instantiation of the candidate generation process, not a fundamental restriction of the framework. We have also discussed this in the limitations section and fully agree that evaluating PEG on broader task types is an important next step. We plan to include additional datasets covering factual, open-ended, and safety-critical tasks in future work.
>
> **W3 and Q1: Ablations on number and quality of discriminators**
>
> Thank you for raising this valuable point regarding the number and quality of discriminators. To investigate these problems, we conduct additional experiments on the ARC challenge set with 5-discriminators (add Gemma-7B [1] and Mistral-7B [2]) and 7-discriminators (further add Ai-Yi-9B [3] and OpenChat-7B [4]). The results highlight the strong impact of PEG on both individual and collective performance. First, as shown in Tables 1 and 2, individual discriminators show consistent improvement after applying PEG. Secondly, in both the 5-discriminator and 7-discriminator settings, our PEG methods consistently outperforms majority vote and ER-D, as shown in Table 3.
>
> However, we observed a slight drop in PEG’s overall performance when moving to more discriminators. We believe this may be due to a mild violation of the Conditional Independence assumption. The two added models (Ai-Yi-9B and OpenChat-7B [3,4]), while strong in general, may not align well with the existing set in terms of signal diversity or informativeness. This observation further reinforces that it’s not just the number of discriminators that matters, but their quality and compatibility with PEG’s theoretical assumptions. The results show if discriminators are chosen in alignment with our underlying assumptions, a larger number of discriminators is not necessarily required to achieve good performance. This suggests that the quality and relevance of the chosen discriminators are more critical than their sheer quantity for the effectiveness of our PEG mechanism.
>
> **Table 1: Accuracy (%) of individual discriminators before and after applying PEG in the 5-discriminator setting.**
>
> |5 Discriminators | OQwen  | deepseek-Qwen | deepseek-Llama2 | Gemma 7B | Mistral 7B |
> |------------------|--------|----------------|------------------|-----------|-------------|
> | After PEG|86.84| 84.27| 83.76| 84.27| 82.74 |
> | Original|82.91|57.18| 59.83| 69.32 | 70.26|
>
> **Table 2: Accuracy (%) of individual discriminators before and after applying PEG in the 7-discriminator setting.**
>
> | 7 Discriminators|OQwen|Qwen|Llama2|Gemma 7B|Mistral 7B|Ai Yi 9B|OpenChat 7B|
> |------------------|--------|--------|--------|-----------|-------------|-----------|---------------|
> | After PEG| 83.85|75.30|73.08|77.61|79.06| 81.62| 82.99|
> | Original| 82.91|57.18|59.83|69.32|70.26| 81.28| 79.06|
>
> **Table 3: Overall accuracy (%) comparison of PEG, Majority Vote, and ER-D under 3-, 5-, and 7-discriminator settings.**
>
> | Setting            | PEG     | Majority Vote | ER-D|
> |--------------------|---------|----------------|---------|
> | 3 Discriminators| 87.01| 70.68| 77.52|
> | 5 Discriminators| 86.75| 71.71| 76.32|
> | 7 Discriminators| 81.97| 76.50| 81.54|
>
> **Q2: Effect of mixed-domain vs. single-domain batches**
>
> We thank the reviewer for pointing out the possibility that each batch may contain a mix of ARC-Challenge and MMLU items.
> We first clarify that our theoretical guarantee of dominant truthfulness relies on the assumption that tasks within each batch are similar. This homogeneity is important because the utility function aggregates feedback across multiple tasks in a batch; if the tasks are too heterogeneous, the computed reward may become ill-posed and fail to provide a meaningful learning signal for effective updates.
> From a theoretical perspective, this assumption ensures that agents can adopt a consistent reporting strategy across tasks within the batch. In contrast, if tasks vary significantly in nature or domain, agents may not be incentivized to follow a stable or truthful strategy. Empirically, we enforce this condition by constructing each batch using items from the same domain within a dataset.
>
> Once again, we appreciate the reviewer's precious time. We are eager to engage in further discussions to clear out any confusion.
>
> [1] Team, Gemma, et al. "Gemma: Open models based on gemini research and technology."
> [2] Albert Q. Jiang, et al. "Mistral 7B."
> [3] Young, Alex, et al. "Yi: Open foundation models by 01. ai."
> [4] Wang, Guan, et al. "Openchat: Advancing open-source language models with mixed-quality data."

---

> > ### Comment · Reviewer_6fgB · 2025-08-02
> >
> > Thank you for the authors' responses. Most of my main concerns have been addressed.
> >
> > Regarding the use of varying discriminators, you mentioned that the drop in PEG’s overall performance with an increasing number of discriminators may be due to a mild violation of the Conditional Independence assumption. Could you elaborate on what types of models are more likely to satisfy this assumption, and how they might be selected in practice?
> >
> > To further verify this hypothesis, it would be helpful if you could replace Ai-Yi-9B and OpenChat-7B with other models that are more aligned with the assumption, and compare performance under the 7-discriminator setting. This would strengthen the claim by providing empirical evidence that performance degradation is indeed related to the violation of the assumption.

---

> ### Author Response · Authors · 2025-08-05
>
> Thank you very much for your thoughtful follow-up and for raising this important point.
> The Conditional Independence assumption ensures that the mutual information–based reward remains informative and does not degenerate. Intuitively, if discriminators are too similar, their outputs become redundant, and observing one adds little value beyond the others. To better satisfy this condition, we recommend using heterogeneous model, as combining models from different families or developers helps reduce dependency. In our experiments, we find that using three diverse discriminators provides a good trade-off between performance and adherence to the CI assumption. This setup is lightweight, stable, and less likely to introduce strong correlations.
>
> We agree that further ablation studies are important for verifying our assumption. To this end, we conducted two additional experiments: (1) using 5 discriminators (OQwen, DeepSeek-Qwen, DeepSeek-LLaMA2, Ai-Yi 9B, and OpenChat 7B), and (2) using 7 discriminators (OQwen, DeepSeek-Qwen, DeepSeek-LLaMA2, Gemma 7B, Mistral 7B, Zephyr 7B, and Phi-2). Results are summarized in Tables 4-6. In Table 6, the rows labeled “5-discriminators (new)” and “7-discriminators (new)” correspond to the additional experiments conducted for our ablation study.
>
> Interestingly, as shown in Table 6, even when combining Ai-Yi and OpenChat, the 5-discriminator setting still achieves strong performance (86.92\%), and PEG consistently improves each discriminator’s accuracy, as shown in Table 4. However, with 7 discriminators, we observe a noticeable performance drop in Table 6. In Table 5, several models (e.g., OQwen) show limited or even negative improvements after PEG. These results support our hypothesis that adding more discriminators does not necessarily improve performance and may even hinder learning. We attribute this to three possible reasons: (1) Adding more models may not provide sufficiently informative or independent signals and can increase the risk of violating the Conditional Independence assumption due to overlapping knowledge or behaviors, making learning less effective; (2) Optimization becomes more difficult when learning from multiple sources with potentially conflicting or low-quality feedback; (3) The inherent capabilities of the discriminator models also play a role: less capable models may contribute limited value or even introduce noise. For example, the newly added Zephyr-7B and Phi-2 are relatively less powerful, which will hinder effective learning.
>
> Based on these observations, we find our method to be robust with different ablations using up to 5 discriminators (as shown in Table 6). Therefore, we propose a practical guideline: start with 3–5 heterogeneous LLMs, which strikes a strong balance between performance, stability, and computational efficiency. Expanding beyond this should be done cautiously and only when additional diversity and independence can be ensured.
>
> **Table 4: Accuracy (%) of individual discriminators before and after applying PEG in the 5-discriminator setting.**
>
> |               | OQwen | Qwen  | Llama2 | Ai Yi 9B | OpenChat 7B |
> |---------------|-------|-------|--------|----------|--------------|
> | After PEG | 87.44 | 83.85 | 83.76  | 86.15    | 85.64        |
> | Original  | 82.91 | 57.18 | 59.83  | 81.28    | 70.26        |
>
> **Table 5: Accuracy (%)  of individual discriminators before and after applying PEG in the 7-discriminator setting.**
>
> |               | OQwen | Qwen  | Llama2 | Gemma 7B | Mistral 7B | Zephyr 7B | Phi-2 |
> |---------------|-------|-------|--------|----------|------------|------------|--------|
> | After PEG | 76.32 | 66.32 | 67.69  | 72.05    | 74.96      | 73.76      | 67.61 |
> | Original  | 82.91 | 57.18 | 59.83  | 69.32    | 70.26      | 68.63      | 61.88 |
>
> **Table 6: Overall accuracy (%) comparison of PEG, Majority Vote, and ER-D under 3-, 5-, and 7-discriminator settings.**
>
> | Setting                 | PEG    | Majority Vote | ER-D |
> |-------------------------|--------|----------------|-----------|
> | 3-discriminators        | 87.01  | 70.68          | 77.52     |
> | 5-discriminators        | 86.75  | 71.71          | 76.32     |
> | **5-discriminators (new)**  | 86.92  | 78.38          | 84.79     |
> | 7-discriminators        | 81.97  | 76.50          | 81.54     |
> |**7-discriminators (new)** | 76.75  | 70.77          | 76.41     |

---

> > ### Comment · Reviewer_6fgB · 2025-08-06
> >
> > Thanks for authors' effort,  I will raise the quality score while maintaining my current rating, with a slight inclination towards acceptance.

---

> > > ### Author Response · Authors · 2025-08-06
> > >
> > > Thank you for your overall positive review of our work! We will incorporate your suggestions into the revised version and would be happy to address any further concerns you may have.

---

### Official Review · Reviewer_KmKz · 2025-06-29

**Clarity:** 3
**Significance:** 3
**Originality:** 3
**Rating:** 5
**Confidence:** 3

**Summary:**

The paper uses a multi-agent system method to address the hallucination problem in language models. The traditional consensus game, whose rewards are based on agreement, falls into the problem of collusion, as the discriminator (the agent judging the outcome) has incentives to agree with the generator (the agent producing the outcome) anyway to receive a higher payment. This paper adopts a peer prediction mechanism, where multiple discriminators are hired to judging the outcome, and their reward are based on comparing each other's answer. In a multi-round dynamic game scenario, the LLM agents converge to a truthful Nash equilibrium with sub-linear regret. The paper also provides experiments to support their theoretical results.

**Questions:**

1. Please address my question in the weaknesses section.

2. What is the covergence rate/speed of the PEG mechanism?

**Ethical Concerns:**

["NO or VERY MINOR ethics concerns only"]

**Final Justification:**

The author's rebuttal addresses my questions. While contrversy remains on what is truthfulness for LLM, I believe the paper is good enough for a broader discussion on NeurIPS. I will maintain my score.

**Limitations:**

Yes.

**Quality:**

4

**Strengths And Weaknesses:**

Strength:

I like this study. It brings me new insight into how peer prediction mechanisms and multi-agent systems can be applied in more trendy areas. The paper has a clear motivation and methodology. While I don't clearly examine the proof, I am more familiar with the DMI mechanism applied and believe this should work at a high level. The algorithmic and experimental results also strengthen the practicality of the study.

Weaknesses:

One thing I would like the authors to discuss more clearly is the notion of "truthfulness" of the LLM agent. In the traditional game theory or mechanism design context, the notion is clear with no ambiguity. But for large language models, this needs more explanation. For example, there can be a problem of calibration. A LM can output "I believe the content is 60% true" with 80% probability and "I believe the content is 40% percent true" with 20% probability. Then, which one is the LM's true belief, and how do we define truthfulness for this LM?

---

> ### Author Rebuttal · Authors · 2025-07-30
>
> We sincerely appreciate your acknowledgment of the quality, significance, and novelty of our work.
> Regarding your concerns, we would like to offer further clarification.
>
> **W1 and Q1: Clarification on the definition of “Truthfulness” in the LLM setting**
>
> We thank the reviewer for raising this important and subtle point. As noted in our submission (Lines 93–102), we adopt the classical notion of incentive compatibility (IC) from mechanism design to define truthfulness. Under IC, an agent is said to be truthful if reporting its private information maximizes its expected utility under the mechanism:
> $u_i(c_i, c_{-i}) \geq u_i(r_i, c_{-i}) \quad \forall r_i \neq c_i,$
> where $c_i \in \\{0,1\\}$ denotes the agent’s true private signal, $r_i$ it's the reported label, and $u_i$ it's the utility based on others' reports $c_{-i}$.
>
> When applied to LLMs, however, the concept of a “true belief” becomes less well-defined. LLM outputs are inherently probabilistic and often miscalibrated, meaning they may not faithfully reflect the model’s internal uncertainty or belief distribution. In our setting, we adopt a practical and operational notion of truthfulness: the report is treated as an output answer rather than a full belief distribution, and we assume the LLM reveals its "true belief" through greedy sampling, without additional exploration. For instance, if the model internally assigns 80\% probability to a correct answer and 20\% to an incorrect one, a truthful response would be the correct label. However, due to hallucinations or the stochastic nature of generation, the actual output may deviate from this internal belief, occasionally resulting in incorrect or noisy responses.
>
> Returning to the reviewer’s example: suppose the LLM outputs “I believe the content is 60\% true” with 80\% probability, and “I believe the content is 40\% true” with 20\% probability. In our framework, we do not treat the textual belief statement itself (e.g., "60\% true") as the truthful signal. Instead, we focus on the correctness label associated with the most probable output—namely, the one with hither probability—as the truthful signal. That is, the model is considered truthful if it reports the option it most strongly believes to be correct. The goal of PEG is to encourage reporting behavior that aligns with the model’s strongest internal belief, ensuring that the output reflects its highest-confidence judgment of correctness.
>
> More broadly, we agree that LLM truthfulness can be meaningfully extended to its belief distribution. In this view, truthfulness becomes closely related to calibration: a truthful LLM should produce outputs that accurately reflect its internal confidence or belief distribution. For example, if the model internally believes the answer is 60\% likely to be correct and 40\% incorrect, then this belief distribution can be treated as its true belief. Miscalibration, a systematic mismatch between internal confidence and the reported output—can result in deviations from incentive-compatible behavior.
> Additionally, some prior works define LLM truthfulness more narrowly, equating it with factual correctness and treating any deviation (e.g., hallucination) as untruthfulness [1]. While this definition captures practical concerns, it does not incorporate ideas from mechanism design or incentive compatibility. The precise definition of LLM truthfulness remains an open question. We will include this discussion in our revised version.
>
> **Q2: Clarification on the covergence rate/speed of the PEG mechanism**
>
> We thank the reviewer for raising this important theoretical question. PEG is implemented via policy mirror descent. Specifically, we use exponentiated gradient updates (akin to mirror descent in the simplex) to iteratively refine the distribution over candidate answers.Therefore, PEG has convergence guarantees in the form of a no-regret property over iterations. Specifically, this implies that the average reward across $T$ steps approaches that of the best fixed policy in hindsight at a rate of $\mathcal{O}\left(\frac{1}{\sqrt{T}}\right)$. This result is consistent with standard findings in prior literature [2,3].
>
> In terms of last-iterate convergence rate, we can observe that PEG typically converges and stabilizes within only 10 iterations in our experiments. However, providing theoretical guarantees for the last iterate remains challenging. Recent work such as [4] analyzes last-iterate linear convergence rates under magnetic mirror descent, but these results are restricted to two-player zero-sum games. We leave the extension of such convergence analysis to the multi-agent peer elicitation games used in PEG as a potential future direction.
>
> Once again, we appreciate the reviewer's precious time. We are eager to engage in further discussions to clear out any confusion.
>
> [1] Azaria, Amos, and Tom Mitchell. "The internal state of an LLM knows when it's lying." arXiv preprint arXiv:2304.13734 (2023).
> [2] Beck, Amir, and Marc Teboulle. "Mirror descent and nonlinear projected subgradient methods for convex optimization." Operations Research Letters 31.3 (2003): 167-175.
> [3] Kiwiel, Krzysztof C. "Proximal minimization methods with generalized Bregman functions." SIAM journal on control and optimization 35.4 (1997): 1142-1168.
> [4] Sokota, Samuel, et al. "A unified approach to reinforcement learning, quantal response equilibria, and two-player zero-sum games." arXiv preprint arXiv:2206.05825 (2022).

---

> > ### Comment · Reviewer_KmKz · 2025-08-03
> >
> > Thank you for your thoughtful rebuttal! I agree taht what means "truthfulness" for LMs is a subtle and subjective question. While I still have concern on whether it is well defined, I believe the idea abd the paper worth to be exposed for a wider range of discussion (given the quality meets the bar). I am happy to maintain my positive score.

---

> > > ### Author Response · Authors · 2025-08-05
> > >
> > > Thank you very much for your thoughtful and positive response! We fully agree that the definition of “truthfulness” is subtle and complex. We’re happy to discuss this further, and we hope that our work can help spark broader conversations around this important topic.

---

### Official Review · Reviewer_FbmF · 2025-07-02

**Clarity:** 3
**Significance:** 3
**Originality:** 4
**Rating:** 5
**Confidence:** 2

**Summary:**

The paper proposes a game-theoretic post-training framework for inducing more truthful responses, by maximizing agreement with a committee of validators. The validators are incentivized to agree with each other. The paper proves three theoretical properties of this framework: (1) that the truthful reporting strategy is optimal; (2) that the updates have sub-linear regret with regard to the best fixed policy; (3) that the final iterate converges to the nash equilibrium. The paper also includes some empirical results on ARC, MMLU, and GPQA, using Qwen models.

**Questions:**

1. The core idea is of consistency among the validators. What prevents the validators from reaching consensus on an incorrect answer?
1. Relatedly, how does the effectiveness of the approach relate to the "hardness" of the prompts? (i.e. are there prompts where the validators initially have high disagreement?)
1. How many validators are needed? Does adding more validators induce a tradeoff between iterations to convergence and the cost of each iteration?
1. Do the implementation use the learning rates from the theory?

**Ethical Concerns:**

["NO or VERY MINOR ethics concerns only"]

**Final Justification:**

I found this paper to be the most thought-provoking of my review batch. As noted in my review, it has several strengths:

- an original, well-motivated, and theoretically-rigorous approach to a very important problem
- strong theoretical guarantees
- well-written paper

I also appreciate the authors' work to resolve my questions and concerns. I recommend that it be accepted.

**Limitations:**

Discussion of limitations is minimal.

**Paper Formatting Concerns:**

bibliography format seems wrong

**Quality:**

4

**Strengths And Weaknesses:**

Strengths

- an original, well-motivated, and theoretically-rigorous approach to a very important problem
- strong theoretical guarantees
- well-written paper

Weaknesses

- a few key questions remain unclear, at least to this reader - see below
- possibly overclaims that the approach works "without ... model fine-tuning" - need more clarity on this
- overclaims about comparisons with larger models (lines 334-342) by focusing on models which are significantly older and are not instruction tuned. Lack of instruction tuning is likely the primary reason that their performance is worse, given that OQwen outperforms these models even *without* PEG.

---

> ### Author Rebuttal · Authors · 2025-07-30
>
> We extend our sincere appreciation for your valuable feedback and suggestions. Regarding your concerns, we would like to offer further clarification.
>
> **W2: Clarification on "without ground-truth labels or model fine-tuning" claim**
>
> Thank you for pointing this out. When we state that PEG operates "without relying on ground-truth labels or model fine-tuning," we mean the following:
>
> First, PEG does not require any  ground-truth labels. Instead, it leverages signals from a group LLM-based discriminators, which serve as referees to provide feedback. The optimization is driven by agreement among these models, rather than comparison to a gold standard.
>
> Second, PEG does not involve any updating on the model parameters. All models—including both the generator and the discriminators—remain completely frozen throughout the process. PEG operates purely by adjusting a distribution over candidate outputs guided by external feedback, making it distinct from supervised fine-tuning or RLHF approaches.
>
> We will revise our claims in the introduction to clarify that PEG relies on peer evaluation rather than ground truth and no model parameter throughout the process.
>
> **W3: Overclaiming due to comparison with older, non-instruction-tuned models**
>
> We thank the reviewer for this valuable comment. We agree that some models used in our comparison are older and not instruction-tuned, which may partially explain their weaker performance. Our intention was not to overclaim, but to maintain consistency with prior work in consensus game [1], where these models were commonly adopted as standard baselines.
> We will remove the text to avoid potential overstatement.
>
> **Q1: What prevents the validators from reaching consensus on an incorrect answer?**
>
> Thank you for the insightful question. PEG relies on consensus among multiple discriminators as a proxy for truthful and high-quality outputs, without assuming that any single discriminator is always correct.
> From a theoretical perspective, our approach is inspired by the peer prediction framework, where heterogeneous agents are incentivized to provide informative and truthful signals under proper reward structure.
> From an empirical perspective, we observe that discriminators often select different correct answers in the early iterations. However, through PEG’s iterative aggregation and update process, these models implicitly “learn from each other” by reinforcing candidates that receive consistent support, while suppressing unstable or weakly supported ones. As a result, the output distribution gradually converges to more coherent and broadly agreed-upon answers, rather than amplifying isolated errors.
>
> **Q2: How does the effectiveness relate to prompt “hardness”? (initially have high disagreements)**
>
> Thank you for raising this insightful question regarding the relationship between prompt "hardness" and the effectiveness of our approach. To address this, we conducted further analysis by categorizing prompts in each dataset into different levels of difficulty based on the initial level of disagreement among the discriminators (validators). This categorization serves as a proxy for prompt difficulty, with higher disagreement indicating greater uncertainty or ambiguity. We report the accuracy for each method across these groups on both ARC-Challenge and MMLU in Table 1.
>
> Our analysis reveals three key findings. First, PEG consistently improves accuracy across all levels of initial disagreement, suggesting that it enables agents to explore diverse judgments and learn from each other to reach better consensus. Second, a counter-intuitive finding is that cases with initial disagreement among discriminators actually result in higher accuracy compared to those with full agreement. We believe this phenomenon occurs because agreement alone does not ensure correctness: all discriminators may still converge on an incorrect answer. In contrast, disagreement introduces diversity, increasing the likelihood that at least one agent is correct, and serves as a useful signal to trigger further exploration. Third, PEG achieves the highest accuracy across all levels compared to other baselines methods. We will include this discussion in our revised version and leave the extension to settings with more than three discriminators for future work.
>
> **Table 1: Accuracy under different initial agreement settings for ARC-Challenge and MMLU datasets.**
>
> | Dataset | Setting |Initial | MI | ER-D| PEG |
> |----------------|--------------|---------|--------|--------|--------|
> | ARC-Challenge  | Agreement    | 64.31   | 73.85  | 72.79  | 84.10  |
> |                | Disagreement | 76.89   | 81.96  | 82.41  | 87.94  |
> | MMLU           | Agreement    | 50.54   | 60.11  | 60.17  | 66.93  |
> |                | Disagreement | 57.96   | 65.19  | 65.02  | 72.20  |
>
> **Q3: How many validators are needed, and is there a tradeoff between iterations to convergence and the cost of each iteration?**
>
> Thank you for raising this important point regarding the number of validators and the associated cost-performance tradeoff. Theoretically, PEG only requires 2 discriminators(validators) to achieve dominant truthfulness. Empirically, We conduct additional experiments on the ARC challenge set with 5-discriminators (add Gemma-7B [2] and Mistral-7B [3]) and 7-discriminators (further add Ai-Yi-9B [4] and OpenChat-7B [5]). The results highlight the strong impact of PEG on both individual and collective performance. First, as shown in Tables 2 and 3, individual discriminators show consistent improvement after applying PEG. Secondly, in both the 5-discriminator and 7-discriminator settings, our PEG methods consistently outperforms majority vote and ER-D, as shown in Table 4.
>
> However, as shown in Table 4, we observe a slight drop in performance when increasing the number of discriminators, even though majority vote and consensus game methods show improvement. We attribute this to a potential violation of the PEG mechanism’s conditional independence assumption, which is critical for its theoretical guarantees and empirical effectiveness. Specifically, the two added models (Ai-Yi-9B and OpenChat-7B [4,5]), while strong in general, may not align well with the existing set in terms of signal diversity or informativeness. This observation suggests that the quality and compatibility of discriminators are more important than their sheer quantity. Additionally, introducing additional discriminators does not increase the number of iterations required for convergence, nor does it add overhead to each iteration. However, as performance becomes slightly more sensitive to the composition of the ensemble, a smaller, carefully curated set of discriminators that adheres to the conditional independence assumption can lead to superior and more stable performance.
>
> **Table 2: Accuracy (%) of individual discriminators before and after applying PEG in the 5-discriminator setting.**
>
> |5 Discriminators | OQwen | deepseek-Qwen| deepseek-Llama2| Gemma 7B | Mistral 7B |
> |------------------|--------|----------------|------------------|-----------|-------------|
> | After PEG|86.84| 84.27| 83.76| 84.27| 82.74 |
> | Original|82.91|57.18| 59.83| 69.32 | 70.26|
>
> **Table 3: Accuracy (%) of individual discriminators before and after applying PEG in the 7-discriminator setting.**
>
> | 7 Discriminators|OQwen|Qwen|Llama2|Gemma 7B|Mistral 7B|Ai Yi 9B|OpenChat 7B|
> |------------------|--------|--------|--------|-----------|-------------|-----------|---------------|
> | After PEG| 83.85|75.30|73.08|77.61|79.06| 81.62| 82.99|
> | Original| 82.91|57.18|59.83|69.32|70.26| 81.28| 79.06|
>
> **Table 4: Overall accuracy (%) comparison of PEG, Majority Vote, and ER-D under 3-, 5-, and 7-discriminator settings.**
>
> | Setting            | PEG     | Majority Vote | ER-D|
> |--------------------|---------|----------------|---------|
> | 3 Discriminators| 87.01%| 70.68%| 77.52%|
> | 5 Discriminators| 86.75%| 71.71%| 76.32%|
> | 7 Discriminators| 81.97%| 76.50% | 81.54%|
>
> **Q4: Clarification on learning rates in the experiments**
>
> Thank you for pointing out the implementation details regarding the learning rate in our experiments. We clarify that the learning rate used in our theoretical analysis is adaptive and depends on a KL divergence term between the initial policy and the optimal policy. However, this term is difficult to estimate in practice. In our empirical studies, we find that the performance of PEG is robust to a range of learning rates. Specifically, we report accuracy results across four datasets using a fixed number of 10 iterations in Table 5. Intuitively, smaller learning rates (e.g., $\leq 0.1$) yield stable performance without compromising convergence speed. In contrast, larger learning rates degrade performance, likely due to the fact that the updates are applied directly to the output distribution, which lies within a bounded space $[0, 1]$. As a result, overly aggressive updates may lead to oscillation or failure to converge. These findings support the practical stability of our approach under reasonable learning rate choices. We will include this experimental results and analysis in our revised version.
>
> **Table 5: Accuracy (%) of PEG with different learning rates across four benchmark datasets.**
>
> | Dataset       | 0.01|0.05|0.1|0.15|0.2|
> |---------------|-------|-------|-------|-------|-------|
> | ARC-Challenge |87.01|87.01|87.01|86.50|80.68|
> | ARC-Easy      |91.82|91.82|91.82|91.65|88.44|
> | MMLU          |70.89|70.88|70.73|68.51|61.06|
> | GPQA          |22.54| 22.54|22.54|22.32|19.42|
>
> [1] Jacob, Athul Paul, et al. "The consensus game: Language model generation via equilibrium search." ICLR
> [2] Team, Gemma, et al. "Gemma: Open models based on gemini research and technology."
> [3]  Albert Q. Jiang, et al.  "Mistral 7B."
> [4] Young, Alex, et al. "Yi: Open foundation models by 01. ai."
> [5] Wang, Guan, et al. "Openchat: Advancing open-source language models with mixed-quality data."

---

> > ### Comment · Reviewer_FbmF · 2025-08-05
> >
> > Many thanks for the detailed response and additional experiments, which clarify my understanding of the submission. It is somewhat surprising that adding validators makes PEG less effective, so it would be worth highlighting this point about the difficulty in maintaining conditional independence with larger sets of validators.
> >
> > My overall assessment remains that this is a strong submission that should be accepted.

---

> > > ### Author Response · Authors · 2025-08-05
> > >
> > > Thank you very much for your thoughtful feedback and for taking the time to engage with our additional experiments. We're glad the clarifications helped, and we agree that the observation about performance degradation with more validators and the associated difficulty in maintaining conditional independence is both surprising and important. We will make sure to highlight this point more clearly in the revised version.
> > >
> > > We sincerely appreciate your positive assessment and suggestions for our submission.

---

### Official Review · Reviewer_tcwC · 2025-07-03

**Clarity:** 2
**Significance:** 2
**Originality:** 3
**Rating:** 4
**Confidence:** 3

**Summary:**

This paper proposes Peer Elicitation Games (PEG), a training-free, game-theoretic framework designed to incentivize truthful outputs from large language models (LLMs) without requiring access to ground-truth labels or fine-tuning. The PEG framework introduces multiple independently instantiated discriminator models that evaluate the output of a generator model. Discriminators are rewarded based on mutual agreement through a determinant-based mutual information scoring mechanism. The authors provide rigorous theoretical guarantees, including incentive compatibility (IC), sublinear regret, and last-iterate convergence to a truthful Nash equilibrium. Empirically, PEG improves factual accuracy across multiple QA benchmarks, showing significant gains even with relatively small models.

**Questions:**

1. In the introduction, the authors highlight PEG’s ability to incentivize truthful reporting without relying on ground-truth labels. Could the authors elaborate more on the disadvantages of using ground-truth labels in this context? For instance, is the goal primarily to reduce human annotation cost, or are there concerns about ground-truth availability, inconsistency, or domain adaptation?
2. Could you add resource analysis, comparing the method with baseline methods?

**Ethical Concerns:**

["NO or VERY MINOR ethics concerns only"]

**Final Justification:**

The paper was not clear enough on the fact that the updates are on distribution only. The authors' explanation makes me understand better the method.

In addition, the author adds more ablation results, e.g. number of iteration, number of discriminator.

**Limitations:**

Yes

**Quality:**

2

**Strengths And Weaknesses:**

Strength:
1. The paper offers a comprehensive set of theoretical guarantees.
2. The method demonstrates consistent performance gains across challenging QA tasks such as ARC-Challenge, MMLU, and GPQA.

Weakness:
1. The workflow of this method is not clear to me, the author claims that this is a training-free method, however it still updates the model using 'silver labels' from majority votes of discriminators? It should be more clear in the paper.
2. It demands high computational and infrastructural resources. For each evaluation task, multiple discriminator models must be instantiated and run across multiple iterations. This includes maintaining several 7B-scale LLMs concurrently and performing multiple rounds of generation-evaluation-update cycles. In some settings, this might outweigh the resource costs of lightweight supervised tuning or distillation.
3. The paper does not provide any quantitative analysis or comparison of computational costs or runtime relative to baselines. Without such comparison, it is difficult to assess the practical viability of PEG in real-world applications, especially in latency- or budget-sensitive settings.
3. The paper does not explore how the number of PEG iterations or number of discriminators affects final performance.

---

> ### Author Rebuttal · Authors · 2025-07-30
>
> Thank you for your insightful feedback on our manuscript. We appreciate your comments and have taken them into consideration to improve our paper. Below, we do our best to address all concerns adequately so that we could receive a better score.
>
> **W1: Clarification on PEG workflow and ''Training-Free" claim**
>
> We appreciate the reviewer’s concern and agree that a clearer explanation of the workflow is necessary. As illustrated in Figure 2 in our submission, the core procedure of PEG consists of four main steps: (1) Given an input question, the generator produces a candidate answer; (2) multiple discriminators independently evaluate this answer, each providing a binary judgment; (3) these judgments are aggregated into a scalar reward signal; (4) a parameterized distribution (policy) over candidate answers is updated using these rewards.
>
> Importantly, the parameters of the generator and discriminators remain entirely frozen throughout the process. All updates occur externally at the level of the output distribution. Thus, we describe PEG as “training-free” in the sense that it requires no gradient-based updates, no backpropagation through model parameters, and no fine-tuning or RLHF.
> Intuitively, the update rule in Equation 4 in our submission increases the probability of outputs that receive higher rewards without requiring access to model internals. Additionally, this approach is consistent with prior training-free paradigms (e.g., [1,2]) that optimize over output distributions rather than over model parameters. We will add these clarifications to Section 2.2 in the revised version to improve clarity.
>
> **W2, W3 \& Q2: Clarification on computational cost and infrastructural resources, resource analysis compared to baselines**
>
> We thank the reviewer for raising this important point. We respectfully clarify that our method does not require maintaining or re-evaluating multiple discriminators across iterations. In each batch, candidate answers are evaluated once by a fixed set of frozen discriminators. The policy update then occurs entirely outside the models at the level of the output distribution.
> As a result, during the iterative process, we simply adjust the probability assigned to candidate outputs: there are no additional generations from the generator, no repeated calls to the discriminators, and no model training (e.g., no gradient updates or parameter tuning). This design ensures computational efficiency and maintains a training-free paradigm throughout the PEG procedure.
>
> To be more specific, as detailed in Appendix C.3, the main computational cost arises during the initial policy extraction, i.e., generating candidate answers from the generator and evaluations from the discriminators. This cost scales linearly with the number of questions. In the subsequent PEG iterations, only lightweight computations (such as reward aggregation and softmax updates) are involved, and no additional infrastructure is required to coordinate multiple discriminators.
> The runtime per iteration of the PEG mechanism for each discriminator is approximately 200 microseconds on average. By comparison, the Consensus Game baseline [1] executes around 40 microseconds per iteration. Notably, PEG processes batches of 8 questions per iteration, enabling efficient parallel updates across multiple inputs.
>
> Importantly, unlike traditional fine-tuning approaches which often require hours to days of compute time, our PEG mechanism is entirely training-free. The full fine-tuning methods generally require thousands of GPU hours and can cost anywhere from $10,000$ to over $35,000$ per run [3]. Even parameter-efficient fine-tuning (PEFT) techniques such as FinLoRA [4] require several hours to days to update model weights. In contrast, PEG operates without modifying model parameters or performing gradient updates, thereby completely eliminating the need for a training phase.
>
> **W4: How the number of PEG iterations or number of discriminators affects final performance**
>
> We thank the reviewer for this valuable question. We conducted additional experiments to study how performance varies with the number of PEG iterations and the number of discriminators.
>
> First, we evaluated the accuracy of PEG with \{10, 20, 30, 40, 50\} iterations across all datasets, using a fixed learning rate of 0.1 and a batch size of 8. From Table 1, PEG converges within the first 10 iterations—subsequent iterations result in only minor adjustments on a few questions, which do not noticeably affect the overall accuracy. We attribute this result to the fact that our updates operate directly in the output policy space rather than modifying model parameters, allowing for faster convergence within a few iterations.
>
> **Table 1: Accuracy (%) of PEG with different numbers of iterations across four benchmark datasets.**
>
> | Dataset        |10| 20| 30| 40| 50|
> |----------------|--------|--------|--------|--------|--------|
> | ARC-Challenge  | 87.01  | 87.01  | 87.01  | 87.01  | 87.01  |
> | ARC-Easy       | 91.78  | 91.82  | 91.78  | 91.78  | 91.78  |
> | MMLU           | 70.78  | 70.81  | 70.81  | 70.79  | 70.81  |
> | GPQA           | 22.54  | 22.54  | 22.54  | 22.54  | 22.54  |
>
> We also conduct additional experiments on the ARC challenge set with 5-discriminators (add Gemma-7B [5] and Mistral-7B [6]) and 7-discriminators (further add Ai-Yi-9B [7] and OpenChat-7B [8]). The results highlight the strong impact of PEG on both individual and collective performance. First, as shown in Tables 2 and 3, individual discriminators show consistent improvement after applying PEG. Secondly, in both the 5-discriminator and 7-discriminator settings, our PEG methods consistently outperforms majority vote and ER-D, as shown in Table 4. However, we observed a slight drop in PEG’s overall performance when adding more discriminators. We believe this may be due to a mild violation of the Conditional Independence assumption. The two added models (Ai-Yi-9B and OpenChat-7B [7,8]), while strong in general, may not align well with the existing set in terms of signal diversity or informativeness. This observation further reinforces that it’s not just the number of discriminators that matters, but their quality and compatibility with PEG’s theoretical assumptions. The results show if discriminators are chosen in alignment with our underlying assumptions, a larger number of discriminators is not necessarily required to achieve good performance. This suggests that the quality and relevance of the chosen discriminators are more critical than their sheer quantity for the effectiveness of our PEG mechanism.
>
> **Table 2: Accuracy (%) of individual discriminators before and after applying PEG in the 5-discriminator setting.**
>
> |5 Discriminators | OQwen  | deepseek-Qwen | deepseek-Llama2 | Gemma 7B | Mistral 7B |
> |------------------|--------|----------------|------------------|-----------|-------------|
> | After PEG|86.84| 84.27| 83.76| 84.27| 82.74 |
> | Original|82.91|57.18| 59.83| 69.32 | 70.26|
>
> **Table 3: Accuracy (%) of individual discriminators before and after applying PEG in the 7-discriminator setting.**
>
> | 7 Discriminators|OQwen|Qwen|Llama2|Gemma 7B|Mistral 7B|Ai Yi 9B|OpenChat 7B|
> |------------------|--------|--------|--------|-----------|-------------|-----------|---------------|
> | After PEG| 83.85|75.30|73.08|77.61|79.06| 81.62| 82.99|
> | Original| 82.91|57.18|59.83|69.32|70.26| 81.28| 79.06|
>
> **Table 4: Overall accuracy (%) comparison of PEG, Majority Vote, and ER-D under 3-, 5-, and 7-discriminator settings.**
>
> | Setting            | PEG     | Majority Vote | ER-D|
> |--------------------|---------|----------------|---------|
> | 3 Discriminators| 87.01| 70.68| 77.52|
> | 5 Discriminators| 86.75| 71.71| 76.32|
> | 7 Discriminators| 81.97| 76.50| 81.54|
>
> **Q1: Clarification on disadvantages of ground truth labels**
>
> Thank you for the thoughtful question. Our motivation for avoiding ground-truth labels stems from these limitations:
>
> First, in many specialized domains such as biomedical, legal, or scientific reasoning, high-quality expert annotations are difficult or infeasible to obtain [9]. This makes ground-truth supervision impractical for real-world deployment and model adaptation in new tasks.
>
> Second, even when annotations are available, they are often unreliable. For open-ended prompts, multiple plausible answers may exist, but ground-truth labels typically assume a single correct one, thus penalizing valid alternatives. In subjective tasks like summarization or ranking, labels are inherently ambiguous, and annotators frequently disagree due to personal preferences, leading to noisy supervision [10].
>
> To address these challenges, PEG replaces reliance on a fixed ground-truth label with feedback aggregated from a panel of random referees. This enables learning signals from each other rather than a single “golden” label, making the framework more scalable and robust.
>
> [1] Jacob, Athul Paul, et al. "The consensus game: Language model generation via equilibrium search." ICLR
> [2] Jacob, Athul Paul, et al. "Modeling strong and human-like gameplay with KL-regularized search." ICML
> [3] Liu, Aixin, et al. "Deepseek-v3 technical report."
> [4] Wang, Dannong, et al. "FinLoRA: Benchmarking LoRA Methods for Fine-Tuning LLMs on Financial Datasets."
> [5] Team, Gemma, et al. "Gemma: Open models based on gemini research and technology."
> [6] Albert Q. Jiang, et al.  "Mistral 7B."
> [7] Young, Alex, et al. "Yi: Open foundation models by 01. ai."
> [8] Wang, Guan, et al. "Openchat: Advancing open-source language models with mixed-quality data."
> [9]  Zhang, Le, et al. "Learning from multiple annotators for medical image segmentation." Pattern Recognition 138 (2023): 109400.
> [10]  Davani, Aida Mostafazadeh, Mark Díaz, and Vinodkumar Prabhakaran. "Dealing with disagreements: Looking beyond the majority vote in subjective annotations." Transactions of the Association for Computational Linguistics 10 (2022): 92-110.

---

> > ### Comment · Reviewer_tcwC · 2025-08-03
> >
> > Thanks for your response. The original paper was not clear enough on the fact that the updates is on probability distribution only, I would suggest to make it more clear in the paper in the next version.
> >
> > I will increase my score.

---

> > > ### Author Response · Authors · 2025-08-05
> > >
> > > Thank you very much for your feedback and for increasing your score. We will make it more explicit that the "policy" in our framework refers to a probability distribution over the output space of the LLM, and that the learning algorithm updates this distribution directly. Thank you again for your helpful suggestion and support.

---

### Note · Authors · 2025-08-12

We thank the reviewers for recognizing the novelty, strong motivation, and theoretical rigor of our work. We also appreciate the constructive feedback which materially improved the paper. Three reviewers now lean towards acceptance, and one has increased the score following the rebuttal.

### Novelty and Contributions

PEG introduces a **training-free, game-theoretical framework for improving LLM outputs via peer elicitation games**. It updates only the **output distribution** without internal gradients, fine-tuning, or RLHF using **determinant-based mutual information score** aggregated from multiple discriminators. This design leverages **peer prediction theory** to incentivize truthful reporting and achieves dominant truthfulness **without requiring ground-truth labels**.

### Additional Experiments and Results

In response to reviewer feedback, we added analyses on **convergence**, **discriminator count and combinations**, and **learning rate**. PEG converges within ~10 iterations, improves individual discriminator accuracy, and outperforms baselines with 3–5 heterogeneous models, remaining robust to learning rate choices. High-disagreement prompts benefit most, showing that diversity can outperform unanimous but incorrect agreement.

We thanks reviewers FbmF and 6fgB for pointing out that adding more discriminators can reduce effectiveness by violating conditional independence; thus, we propose a **practical guideline**: start with 3–5 diverse LLMs for a balance between performance and computational efficiency, expanding cautiously only when additional diversity and independence are ensured.

### Clarifications

Addressing reviewers tcwC and 6fgB: we clarified **computational efficiency** (runtime analysis) and **theoretical assumptions** (binary labels and conditional independence), as well as the disadvantages of operating with ground-truth labels. For reviewer KmKz, we clarified our **definition of truthfulness**: the model reports the answer it most strongly believes correct, aligning its output with its highest-confidence internal belief.

### Overall Impact

**PEG provides a principled and efficient framework for producing more truthful LLM outputs without relying on labels or model retraining. It achieves higher accuracy, fast convergence, and robustness across diverse conditions, making it a practical and theoretically grounded solution for improving LLM truthfulness and outputs quality.**

---

### Decision · Program_Chairs · 2025-09-17

**Decision:**

Accept (poster)

**Comment:**

The paper introduces Peer Elicitation Games (PEG), a training-free, game-theoretic framework for improving truthfulness in LLM outputs.
All reviewers see the paper as novel, theoretically rigorous, and relevant. After rebuttals and discussions, consensus is positive.

Strengths of the paper highlighted are:

- Novelty & theory: Original application of peer prediction to LLM truthfulness; rigorous proofs of IC, regret bounds, and convergence.
- Training-free paradigm: PEG updates only the output distribution, requiring no gradient updates or retraining.
- Empirical evidence: Improves accuracy across difficult QA benchmarks using small open-source models.

Of course the paper also comes with some weaknesses the reviewers pointed out, including:
- Computational Cost: Initially unclear; rebuttal clarified that PEG’s iteration cost is lightweight once candidates are generated, but infrastructure overhead still exists.
- Truthfulness Definition: Reviewer KmKz raised concerns about defining “truthfulness” for probabilistic LLMs; authors clarified they adopt a pragmatic definition (reporting the model’s most likely belief). This remains an open conceptual issue.

This submission offers a well-motivated, and theoretically solid framework for eliciting truthfulness from LLMs. While some assumptions (binary labels, conditional independence) limit the generality of the guarantees, the combination of theoretical rigor and empirical improvements makes PEG a significant contribution. Therefore based on the reviewers' feedbacks, I would recommend acceptance.